# Making a game out of exploration-exploitation

## Abstract

What is the best way for an agent to balance exploration with exploitation? In this paper we suggest an answer to this question that treats exploration and exploitation as independent players competing to maximize a joint objective. Through theory and simulations we show how a "game" played between two deterministic policies, one maximizing intrinsic curiosity and one maximizing extrinsic environmental rewards, yields a simple maximum value solution over both policies. The key assumption that allows for this is our assumption that curiosity and reward seeking are equally valuable on evolutionary terms. We start by developing an axiomatic approach to defining information value that generalizes past approaches, while simplifying our ability to estimate such value in both artificial and biological memory systems. We then show how our deterministic solution performs at least as well as standard stochastic explore-exploit algorithms, but has the added benefit of being far more resilient to deceptive rewards (i.e., local minima), more efficient in high-dimensional action contexts, and robust to hyperparameter choices. Thus, the solution to our version of the gamified exploration-exploitation problem can be summarized by a simple heuristic: when the expected value of information is more than the expected value of rewards, be curious, otherwise seek rewards.

## 1 Introduction

One of the most fundamental questions in reinforcement learning is determining when to explore searching for new information, and when to exploit existing knowledge to maximize reward. Or as stated in Sutton & Barto (2018), "The agent has to exploit what it has already experienced in order to obtain reward, but it also has to explore in order to make better action selections in the future. The dilemma is that neither exploration nor exploitation can be pursued exclusively without failing at the task". We argue the dilemma is not singular, but often contains two decisions. First is the decision to explore at all versus simply maximizing expected rewards. Second is the choice of which action to explore if exploration is chosen. The optimality of both decisions is often uncertain. One common strategy, $\epsilon$-greedy Sutton & Barto (2018), handles both uncertainties by randomly making both decisions. A more directed approach is to augment reward values from the environment with intrinsic rewards or motivations. A variety of intrinsic motivation strategies are available in the literature. Examples include, novelty signals Kakade & Dayan (2002), action counts Bellemare et al. (2016), information gain Friston et al. (2017), error maps Thrun (1992), curiosity Schmidhuber (1991) and learning progress Kaplan & Oudeyer (2007). Another approach altogether is to use pure exploration and deterministically and sequentially tries all actions for a fixed number of steps, selecting the most valuable option only at the end of exploration Brafman & Tennenholtz (2002); Strehl et al.; Kearns & Singh (2002).

Randomly resolving the dilemma as in $\epsilon$-greedy is effective and in fact a common solution in the literature. It is however inefficient and can struggle in continuous and high-dimensional action/state spaces (Sutton & Barto, 2018). Pure exploration can often be guaranteed to find the optimal/most rewarding action in bandit settings but is likewise inefficient with large problems (Brafman & Tennenholtz, 2002; Seldin et al.). Perhaps more important though is the fact that pure exploration ignores rewards value during its search completely. Whereas exploration augmented by intrinsic rewards will often fair better for larger problems, but it requires parameter tuning and so can be computationally expensive to arrive at optimal explore-exploit strategies.

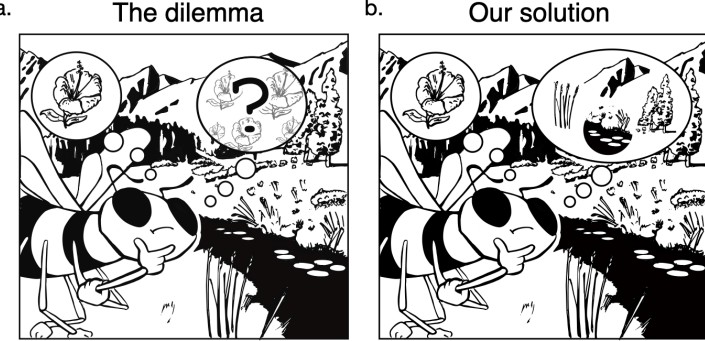

Figure 1: Two views of the exploration-exploration question. **a**. The classic dilemma: either exploit an action with a known reward by returning to the best flower or explore other flowers on the chance they will return a better outcome. The central challenge here is that the exploration of other flowers is innately uncertain in terms of the pollen collected, the extrinsic reward. **b**. Our alternative: an agent can have two goals: either exploit the reward from the best flower (action), *or* explore to maximize pure learning with a curious search of the environment. In the dilemma (a.) the expected value of exploration is uncertain. Whereas to solve our alternative view (b.), we set information and reward value on equal terms, approximate their expected values, and derive a deterministic, optimal value rule to choose between exploration and exploitation. In other words, our alternative transforms the dilemma into a simple greedy decision. *Artist credit*: Richard Grant.

In this paper we are interested in simplifying the use of intrinsic rewards, or simplifying really the use of curiosity. To do this we wonder if exploration should consider reward value at all? Or, to put it another way, should exploration in reinforcement learning consider reward maximization at all? Or instead, can exploration be considered as pure exploration motivated by learning progress. We use theory and simulations to show exploration need not consider reward value, and that the character of the standard explore-exploit question simplifies if we treat environmental rewards and intrinsic information value as equally important (Inglis et al., 2001) objectives. We show that if they are equally important, then agents can choose the larger expected value between them and know with certainty they have gotten the "best" return available. This seems to profoundly simplify optimal, or near optimal, explore-exploit solutions, which normally require expensive action sampling or computationally expensive Bayesian planning.

To solve the dilemma, we put strategies for reward collection and learning progress/curiosity in competition with one another as illustrated in Figure 1b. We find it fruitful to view this competition as a maximum value "game", played out in an agent's "mind". Here two policies compete for control of the agent's decisions, but share a common objective of maximizing the joint value of external rewards and intrinsic information value.

Animal behavior was a key motivator in the development of reinforcement learning theory Sutton & Barto (2018). Our focus on curiosity is in very much the same spirit. Animals in the natural world often strongly prefer to satisfy curiosity over receiving tangible food or water rewards from their environment (food, water, etc). They do so even when that information is costly in terms of time, energy, or both (Song et al., 2019; Wang & Hayden, 2019; Taylor, 1975; Singh, 1970). This fact stands in contrast to the common assumption in reinforcement learning that environmental rewards are the primary aim. A reader accustomed to thinking of animals as externally motivated by rewards might wonder why animals should show such curious preferences? The answer is simple. Curiosity is a profoundly useful strategy which appears universally across the animal kingdom, as reviewed in Loewenstein (1994); Kidd & Hayden (2015). As reviewed below, this exploration strategy does not appear directly linked to reward collection, but is instead essential for agent survival, causal learning, language learning (in humans) injury recovery and avoidance, adaptation to non-stationary environments, and the building of world models, strategic planning, and the avoidance of deceptive rewards.

Simulation experiments using artificial curiosity show that it leads to the building of intuitive physics (Laversanne-Finot et al., 2018), and may be key to understanding causality (Sontakke et al., 2020). It

can help ensure that an agent has generalizable world models (Kim et al., 2020; Wang & Hayden, 2020). It can drive imagination, play, and creativity (Schmidhuber, 2008; Auersperg, 2015; Haber et al., 2018; Friston et al., 2017). It can lead to the discovery of, and creation of, knowledge (Berlyne, 1954; Loewenstein, 1994; Zhou et al., 2020). It can ensure deceptive value functions, those whose rewards appear suboptimal early in learning but are optimal in the long term, are overcome during learning (Fister et al., 2019; Pathak et al., 2019) (an especially important point that we revisit in our experiments later on). Curiosity leads to the discovery of changes in the environment (Sumner et al., 2019), and ensures there are robust action policies to respond to these changes (Dubey & Griffiths, 2020). Curiosity can help in the recovery from injury (Cully et al., 2015). It can drive language learning (Colas et al., 2020). It can aid in creating a reservoir of action policies Groth et al. (2021). It is critical to evolution and the process of cognitive development (Oudeyer & Smith, 2016; Gopnik, 2020).

In other words, curiosity has diverse benefits that makes it useful to agents, both artificial and biological. In fact, on the basis of this universality we assert that curiosity is sufficient for all exploration in reinforcement learning (Groth et al., 2021; Fister et al., 2019; Inglis et al., 2001). We further hypothesize that the dual objectives of maximizing external rewards and information are on equal terms evolutionarily speaking: the latter aids survival in understanding of changing environments, while the former aids in more immediate ways by supplying food, water, and other biological requirements.

However despite curiosity being a powerful method to build world models, avoid local minima, discover causation, there remain two open problems in curiosity research we felt needed to be addressed before curiosity could be fully incorporated into reinforcement learning theory in a general way:

1. The existing definitions of curiosity and information value are either task dependent, memory specific, or specific to the learning rule. Or all of the above. For example, in a recent review Oudeyer ((Oudeyer, 2007) created a formal "typology" of computational theories of intrinsic motivation, focusing on the creating classifications of intrinsic motivation during learning based on in part on their motivations or goals. He stops short though of defining a general metric, which is what we offer here.

2. Curiosity algorithms in artificial systems suffer from a "white noise" problem where they become fixated on "useless" high entropy features in the environment (Pathak et al., 2017; Kim et al., 2020).

## 1.1 Contributions.

In this paper we develop:

- An axiomatic account of information value based on a generalization of learning progress Kaplan & Oudeyer (2007). Our account of information value is based on only the recursive dynamics of memory, independent of the learning rule or objective. This account generalizes over existing count-based methods, information gain, novelty signals, and all previous methods to measure learning progress. To value information we do not require understanding the motivations for learning, the meaning of learning, or the means of learning. We aim to embody curiosity as a drive towards "learning for learning's sake" or very general notion of learning progress Kaplan & Oudeyer (2007).

- We establish a Bellman optimality for our account of information value. This is necessary to incorporate information value into classic reinforcement learning, which is itself based on the Bellman optimality. Our result is based on a novel idea of targeted forgetting that removes the typical requirement of keeping the entire learning history intact in order to establish optimal substructure for information.

- We explore two strong assumptions for information value. First, we suggest that maximizing information value and maximizing reward value are independent but equally important objectives in reinforcement learning. Second, we suggest curiosity is a sufficient objective for all exploration in reinforcement learning. This second assumption is based on the universality of curiosity in the natural world (as reviewed above).

- We use these assumptions to prove there exists a "meta-Bellman" rule to choose between pure deterministic policies for exploration (curiosity) and exploitation (rewards).

- We simplify this meta-policy further to implement a win-stay lose-switch rule (Nowak & Sigmund, 1993; Bonawitz et al., 2014; Worthy & Todd Maddox, 2014). In making a game out of the explore-exploit problem we also hope to make this game as simple to play and therefore as simple to compute as possible.

- We study this win-stay lose-switch (WSLS) rule in a number of computer simulations. We show it performs equivalently as state-of-the-art methods for resolving explore-exploit in classic bandit environments but also show that it is highly robust to deceptive rewards, and to hyperparameter selection. In other words, we show curiosity-driven exploration can perform as well as reward-motivated exploration and provide additional benefits.

- We also establish that the "white noise" problem found in many artificial curiosity algorithms is naturally eliminated in our explore-exploit game without the need to constrain the objective of curiosity itself, as is otherwise common (Pathak et al., 2017; Kim et al., 2020).

## 1.2 Related work

We are not the first to propose a unity for curiosity and information value. For example, Friston et al. (2017) asserted that agents *must* minimize the entropy of their sensations, and so proposed to unify curiosity as a means to minimize uncertainty via active sampling and Bayesian reasoning. In other words, they prescribe an objective, representation, and mechanism for curiosity. This is just one of many prior approaches to curiosity and exploration (Berlyne, 1954; Loewenstein, 1994; Friston et al., 2017; Lydon-Staley et al., 2021; Oudeyer, 2018; Schmidhuber, 2019; Kim et al., 2020; Schmidhuber, 2008; Wilson et al., 2021). Our aim is not to debate the merits of any one of these. Instead we aim to develop a mathematical account that generalizes across *all* of them. Specifically, we develop a perspective of curiosity that does not require knowing the objective of curiosity itself (only the value of information), does not assume there is only one objective between rewards and information, does not assume a specific kind of representation, and does not prescribe mechanisms of learning beyond normative reinforcement learning. By not requiring these details we can achieve a strong generality, one that we hope is suited to artificial and biological agents alike.

Our approach to information value is built on ideas taken from the *learning progress* literature (Kaplan & Oudeyer, 2007; Lopes et al., 2012; Baranes & Oudeyer, 2013; Ten et al., 2021). The key point of difference between our approach and this prior work is that we do not include in our value metric the learning objective or loss function itself, as all the learning progress metrics that we are aware of do, either directly or indirectly (Oudeyer, 2007; Graves et al., 2017; Oudeyer, 2018; Haber et al., 2018). Inclusion of the learning objective or loss function undercuts our desire to develop a general formalism for learning progress, information value, and for curiosity. As it turns out, removing these details may also be convenient in practice. We can measure information value directly, given only observations, for most any kind of memory system.

We will go on to formalize environmental reward collection and then contrast it to a formal definition of information value collection and curiosity. This might lead the reader to wonder if our information value is itself a reward or, to be consistent with past work, should we just denote information value as an "intrinsic reward" and contrast it with "extrinsic rewards"? This is after all a fairly common distinction. So long as we are consistent in our definitions and use of terms our meaning should be clear. But we have refrained from these distinctions for two reasons. One, we mean something specific by information value and do not wish to conflate it with the more general intrinsic reward label. Two, the standard theoretical frameworks for reinforcement learning makes it seem as if extrinsic (i.e., from the environment) and intrinsic (i.e., from the agent) value are quite independent. However, the circuits for environmental reward valuation in real animals depend in part on satiety, reward history, and other preferences intrinsic to the animal. This suggests to us the split between intrinsic and extrinsic is less clear in practice than it is in theory.

As far as we are aware, we are the first to factor exploration-exploitation into a game between curiosity and reward. Many others have however separated out explore-exploit into independent parts. In one recent example of explore-exploit separation, (Colas et al.) suggested a fixed-length curious search ought to proceed

with reinforcement learning in order to build up useful task-agnostic representations. Groth et al (Groth et al., 2021) suggested saving task-agnostic curiosity policies as a basis for learning task-specific reinforcement learning models. Zhang et al (Zhang et al., 2019) derived a randomized approach which alternated between exploration, using a specific representation, and exploitative reinforcement learning. In an effort to scale another learning algorithm, Clune et al experimented with a mixture of two policies for novelty-seeking and task performance (Colas et al., 2020). However the closest prior analog to our work seems to be found in an article from organizational psychology. In their work, Gupta et al (Gupta et al., 2006) argue it may be fruitful for business organizations to switch between periods of pure curious exploration for product ideas, and pure periods of exploitation of existing products.

## 2  Theory

To describe the environment we use a discrete time Markov decision process, $\mathcal{X}_t = (\mathcal{S}, \mathcal{A}, \mathcal{T}, \mathcal{R})$. States are real valued vectors $S$ from a finite set $\mathcal{S}$ of size $n$. Actions $A$ are from the finite real set $\mathcal{A}$ of size $k$. We consider a set of policies that consist of a sequence of functions, $\pi = \{\pi_t, \pi_{t+1}, \ldots, \pi_{T-1}\}$. For any step $t$, actions are generated from states by policies, either deterministic $A = \pi(S)$ or random $A \sim \pi(S|A)$. In our presentation we drop the indexing notation on our policies, using simply $\pi$ to refer to the sequence as a whole. Rewards are single valued non-negative real numbers, $R^+$, generated by a reward function $R \sim \mathcal{R}(S|A, t)$. Transitions from state $S$ to new state $S'$ are caused by a stochastic transition function, $S' \sim \mathcal{T}(S|A, t)$. We leave open the possibility both $\mathcal{T}$ and $\mathcal{R}$ may be time-varying. In general we use the $=$ to denote assignment, in contrast to $\sim$ which we use to denote taking random samples from some distribution. The standalone $|$ operator is used to denote conditional probability. An asterisk is used to denote optimal value policies, $\pi^*$.

### 2.1  Reward collection

We begin by restating the standard reinforcement learning problem of maximizing rewards (Sutton & Barto, 2018). We do this to set the stage for our derivation for information value and also to define the reward maximizing policy $\pi_R$ used in our final equations (For example, Eq.7). To do this effectively, however, we need to clarify the following nomenclature. Here we will refer to the intrinsic motivation of any external stimulus as simply information value, though noting that it is just as capable of reinforcing learned behavior as any extrinsic reward might be. We will then use reward as a specific shorthand for value that is established and given by the external environment.

We consider an agent that interacts with an environment over a sequence of states, actions and rewards. The agent's goal is to select actions in a way that maximizes the total reward, $R$, collected. How should an agent do this? The standard way of approaching exploration in reinforcement learning is to assume, "The agent must try a variety of actions and progressively favor those that appear to be the most rewarding" (Sutton & Barto, 2018). So given a policy $\pi_R$, the value function in standard reinforcement learning is given by Eq. 1. This is the term we use to maximize reward collection.

$$V_{\pi_R}(S) = \mathbb{E}_{\pi_R}\Big[ \sum_{t=0}^{T} R_t \mid S_t = S \Big] \tag{1}$$

But this equation is hard to use in practice because it requires that we integrate over all time, $T$. The Bellman equation (Eq. 2) is a desirable simplification because it reduces the entire action sequence in Eq. 1 into two (recursive) steps, $t$ and $t+1$. This rule can then be recursively applied. The practical problem we are left with is finding a reliable estimate of $V_{\pi_R^*}(S_{t+1})$ (Eq. 2). This is a problem we return to further on.

$$\begin{aligned} V_{\pi_R^*}(S) &= \max_{\pi_R} V_{\pi_R}(S) \\ &= \max_{A \in \mathbb{A}} \mathbb{E}\Big[ R_t + V_{\pi_R^*}(S_{t+1}) \mid S_t = S, \ A_t = A \Big] \end{aligned} \tag{2}$$

Here we have focused on finite time horizons and we do not discount the rewards over time as is typically done in classic reinforcement learning. Both are done simply for simplicity. The basic approach should generalize to continuous spaces and discounted but infinite time horizons (Bertsekas, 2017)

## 2.2 Information collection

Our aim is to integrate independent value functions for reward collection and information collection into a theoretical whole. This means for us two things, we need to derive a Bellman equation for information collection inline with those equations used for reward collection. We also feel it ideal to introduce a notion of information value that is as general as possible. The former happens in this section. The latter we do axiomatically in the next section. In using axioms we are hoping to build a notion of information value that is nearly as general as information theory itself.

We next consider an agent that interacts with the environment over the same space as in reward collection, but whose goal is now to select actions in a way that maximizes information value (Schmidhuber, 1991; Oudeyer, 2018; Burda et al., 2018; Zhang & Yu, 2013; de Abril & Kanai, 2018; Zhou et al., 2020; Schwartenbeck et al., 2019; Wilson et al., 2014; Lehman & Stanley, 2011a; Velez & Clune, 2014). How should an agent do this? To find this answer we face two more questions: 1) how should information be valued?, 2) and, like for reward collection, is there a Bellman solution for information value?

If $E$ represents information value, then the Bellman solution we want for $E$ is given by Eq. 3.

$$V_{\pi_E^*}(S) = \max_{A \in \mathbb{A}} \mathbb{E}\Big[E_t + V_{\pi_E^*}(S_{t+1}) \mid S_t = S, \ A_t = A\Big] \tag{3}$$

We assume that maximizing $E$ also maximizes curiosity. We refer to this greedy and deterministic approach to curiosity as $E$-exploration, or $E$-explore for short.

In the following paragraphs we define $E$ and prove this definition satisfies the Bellman solution shown in Eq.3. In doing this we will *not* limit ourselves to Markovian ideas of learning and memory.

We define memory $\mathcal{M}$ as a vector of size $p$ embedded in a real-valued space that is closed and bounded. This idea maps to a range of physical examples of memory, including firing rates of neurons in a population (Wang, 2021), strength differences between synapses (Stokes, 2015), or calcium dynamics (Higgins et al., 2014), as well as to the weight vectors common in deep reinforcement learning (Arulkumaran et al., 2017).

A learning function is then any function $f$ that maps observations of the Markov space, $\mathcal{X}$, into memory $\mathcal{M}$. This idea can be expressed using recursive notation, denoted by $\mathcal{M} \leftarrow f(\mathcal{X}, \mathcal{M})$. Though we will more often use time indexing $\mathcal{M}_t$ to denote the updated memory instead. In this notation the difference between and two memories $\mathcal{M}_{t-\tau}$ and $\mathcal{M}_t$ is denoted by $\Delta\mathcal{M}$, for some finite time difference $\tau > 0$. We will also need to define a forgetting function, $f'(\mathcal{X}_{t-1}, \ \mathcal{M}_t) \rightarrow \mathcal{M}_{t-1}$ which is important later on when establishing our novel Bellman result. As an auxiliary assumption we assume $f$ has been selected by evolution to be formally learnable (Valiant, 1984).

A more traditional kind of learning function could be something like $f_{\mathcal{M}}(\mathcal{X}) = \hat{y}$ and we would be concerned with minimizing the loss between a target objective $y$ and the estimate $\hat{y}$, based on *parameters* $\mathcal{M}$. Indeed, such a function will be in operation "behind the scenes". In this paper we are not concerned with the objective $y$ and the absolute error of the loss function, but only with the memory dynamics themselves and their equilibrium points.

### 2.2.1 Axioms for information value

We next define a new metric of information value, where information value is dependent on learning and *learning progress* (Oudeyer, 2007). We will define a semi-metric that is general, practical, and simple to measure for nearly any memory system. That is, we do not assume the learning algorithm or loss function is even known or at all important in assessing the value of information value. We only assume that we have a memory system–or a parameter space, which we treat as just one sort of memory–whose learning dynamics we can measure. We are, in other words, concerned only with how much was learned and not how it was

learned. We reason that the value of any piece of information depends entirely on how much it changes in memory. We formalize this idea with a formal set of axioms.

**Axiom 1** (Axiom of Memory). *The value of $E$ depends only on the change in memory, $\Delta\mathcal{M}$.*

Mote that a corollary of Axiom 1 is that an observation that does not change memory has no value. So if $\sum_{i=1}^{p} |\Delta M_i| = 0$ then $E = 0$.

**Axiom 2** (Axiom of Specificity). *If all $i$ elements $\Delta M_i$ in $\Delta\mathcal{M}$ are equal, then $E$ takes on a minimal value.*

In other words, information about $\mathcal{X}$ that is uniformly encoded in the space of $\mathcal{M}$ is non-specific, or, in a sense, maximum entropy. We do not mean that when all $i$ elements $\Delta M_i$ in $\Delta\mathcal{M}$ are equal, then $E = 0$ and so $E$ is minimized in an absolute sense. Instead we think of this axiom as an "inverse of entropy" axiom. Entropy is maximized under a uniform distribution, and so we reason information value is minimized under the same condition. Learned information evenly in memory cannot induce any specific inductive bias on behavior, and is therefore regarded as the least valuable information possible (see also(Mitchell, 1980)). As an example, if say the two element memory difference with the $\sum \Delta\mathcal{M} = 0.25$ but is also the vector $[0.125, 0.125]$ then this vector will have the smallest $E$ compared to any other vector that satisfies that same sum.

**Axiom 3** (Axiom of Scholarship). *Learning has an innately positive value. Therefore, $E \geq 0$.*

Thus, information value restricted to be positive is defined by some absolute change in $\mathcal{M}$. We believe that this axiom holds in all cases even if for some particular learning instance information has negative consequences for the agent. That is, we choose to regard learning as "good" even in the face of "bad" consequences.

**Axiom 4** (Axiom of Equilibrium). *Given some observation $\mathcal{X}$, $E$ will decrease below some finite threshold $\eta > 0$ in a finite time horizon $T$.*

Putting Axioms 1-4 together, we have a geometric perspective of information value. To make this abstract notion more concrete in Figure 2a we show an example of $E$ defined in a memory space where $\mathcal{M}_i$ and $\mathcal{M}_j$ reflect two dimensions of a memory space–a simple two unit system with reciprocally connected neurons labeled $A$ and $B$ (inset in Figure 2a). In this example, $\mathcal{M}_i$ reflects the real valued "weight" of $A \rightarrow B$ and $\mathcal{M}_j$ would reflect the weight of $B \rightarrow A$. At any time $t$ the weights $\mathcal{M}_i$ and $\mathcal{M}_j$ have specific values (what those values are is not important for this example). Now, let's consider a potential observation, or experience, that the system can have at time $t + \tau$. Due to some hidden learning process we change the weights on $A$ and $B$. The (scalar) distance between the values of $\mathcal{M}_i$ and $\mathcal{M}_j$ from time $t$ to time $t + \tau$ directly reflects the information value, $E$. The larger the scalar distance, the larger the $E$, and thus more value.

In Figure 2b we depict different trajectories for dramatically different patterns in learning. In the top panel the overall dynamics are consistent with changes in $E$ by our definition in Axiom 4. In contrast, Figure 2c shows a pattern of learning-related changes in $E$ that would be inconsistent with our requirement in Axiom 4. These notions of consistency follow from the fact we only ensure memory dynamics converge, and so they become self-consistent. Another equally good term could be contracting, as in $E$ is contracting semi-metric. What we do not do and cannot do is to ensure that errors are minimized overall, and therefore learning and memory is "correct". That is, while we require the dynamics to reach equilibrium in finite time, we do not require that the unknown and unstated error between the memory and the Markov observation is minimized. That is, we make no assumptions about convexity or convergence. It turns out that this simpler notion of self-consistency is sufficient for a useful curious search, at least in our simulations (see below).

It also follows from Axiom 3 and 4 that curiosity-driven exploration will visit every state in $\mathcal{S}$ at least once in a finite time $T$, assuming $E_0 > 0$. This follows from the fact that once $E - \eta = 0$, exploration of the corresponding observation $\mathcal{X}$ will cease. This implies that each state will eventually become "unavailable" forcing any E-exploration algorithm to visit some other state. This process will repeat until all states have been visited. In addition, as we stop exploring when $E - \eta = 0$, only those states in which there is more to be learned will be revisited. This is, in other words, a maximum entropy exploration policy Hazan et al. (2019). This, along with the deterministic nature of our policy, ensures perfectly efficient exploration in terms of learning progress.

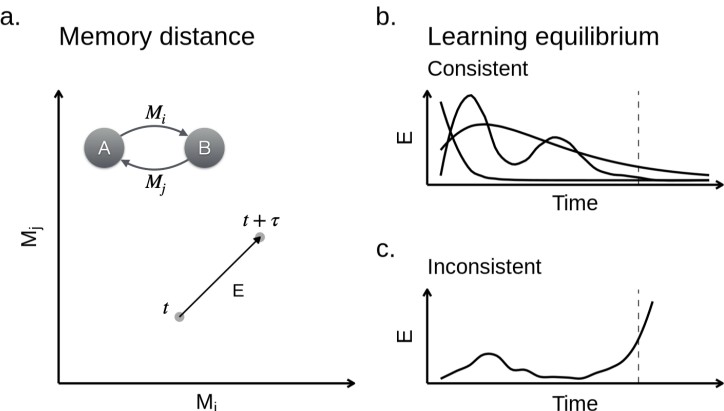

Figure 2: Information value. **a.** An example of information value in a two dimensional memory system. For example the directed weights from unit $A$ to unit $B$ and vice versa (inset model). The information value of an observation depends on the distance memory "moves" during learning, from time point $t$ to $t+\tau$, denoted here by $E$. We depict memory distance as euclidean space, but this is one of many possible ways to realize $E$. **b** Examples of learning dynamics in time, made over a series of observations that are not shown. If information value becomes decelerating in finite time bound, then we say learning is consistent with Axioms 1-4. **c** If learning does not decelerate, then it is said to be inconsistent. The finite bound is represented here by a dotted line.

Our axioms for $E$ are useful in three unique ways. First, they provide a measure of information value *without* needing to know the exact learning algorithm(s) being used. This leads to the second useful property of our axioms: if we want to measure any agent's representation of information value, we need only record differences in dynamics of their memory circuits. There is little need to decode, or interpret. Finally, and perhaps most importantly, these axioms properly generalize prior attempts to formalize curiosity. In other words, prior definitions of information value for curious exploration all fall under the definition we provide here.

There are range of mathematical formulations that satisfy our axioms for the semi-metric, $E = d(\mathcal{M}_{t-1}, \mathcal{M}_t)$, for some measuring distance function $d$. Without loss of generality here we will make a *working definition* suitable for our multi-armed bandit simulations featuring stochastic rewards. Namely we use the Kullback-Leibler divergence (KL), a common and classic measure of information gain is the KL divergence MacKay (2003) where we restrict $\mathcal{M}$ to be a memory for the probability of outcomes, $\mathcal{X}$.

$$E = d(\mathcal{M}_t, \mathcal{M}_{t-1}) = \text{KL}(\mathcal{M}_t | \mathcal{M}_{t-1}) \tag{4}$$

It is important to note however that the above definition is one of several options that can be freely chosen for $E$ depending on the details of $\mathcal{M}$ and $f$. For example, the lp-norm of the difference between memories of vectors of real numbers, $\|\Delta\mathcal{M}\|_p$, satisfies our axioms. As does the norm of gradient, $\|\nabla\mathcal{M}\|_p$, a fact that is especially useful for studying curiosity and information value in differential programs and deep learning.

### 2.2.2 The importance of boredom

Of course, curiosity has its limits. To avoid the white noise problem (Kim et al., 2020), other useless minutia (Pathak et al., 2017), and to stop exploration efficiently we rely on the threshold term, $\eta \geq 0$ (Ax. 4). We treat $\eta$ as synonymous with boredom as defined elsewhere (Hill & Perkins, 1985; Schmidhuber, 1991; Bench & Lench, 2013). In other words, we hypothesize that boredom is an adaptive parameter tuned to fit the environment (Geana & Daw, 2016). Specifically, we use boredom to ignore arbitrarily small amounts of information value, by requiring exploration of some observation $\mathcal{X}$ to cease once $E \leq \eta$ for that $\mathcal{X}$.

### 2.2.3  A Bellman solution for maximizing information value

A common way to arrive at the Bellman solution is to rely on a Markov decision space. This is a problem for our definition of memory as it has path dependence, which violates the Markov property assumption. To get around this, we prove that exact forgetting of the last observation is another way to find a Bellman solution.

To find a dynamic programming solution based on the Bellman equation (Eq 3) we must prove our memory $\mathcal{M}$ has optimal substructure. This is because the normal route, which assumes the problem rests in a Markov Space, is closed to us. By optimal substructure we mean that the process of learning in $\mathcal{M}$ can be partitioned into a collection, or series of memories, each of which is itself a solution. That is, it lets us find a kind of recursive structure in memory learning, which is what we need to apply the Bellman. Proving optimal substructure also allows for simple proof by induction (Roughgarden, 2019), a property we will take advantage of.

**Theorem 1** (Optimal substructure). *Let $\mathcal{X}_t$, $\mathcal{M}$, $f$, $\pi_E$ and $\mathcal{T}$ be given. Assuming transition function $\mathcal{T}$ is deterministic, if $V^*_{\pi_E}$ is the optimal information value given by policy $\pi_E$, a memory $\mathcal{M}_{\sqcup-\infty}$ has optimal substructure if the the last observation $\mathcal{X}_t$ can be removed from $\mathcal{M}_{t-1}$, by $\mathcal{M}_t = f'(\mathcal{M}_{t-1}, \mathcal{X}_t)$ such that the resulting value $V^*_{t-1} = V^*_t - E_t$ is also optimal.*

**Proof:** *Given a known optimal value $V^*$ given by $\pi_E$ we assume for the sake of contradiction there also exists an alternative policy $\hat{\pi}_E \neq \pi_E$ that gives a memory $\hat{M}_{t-1} \neq M_{t-1}$ and for which $\hat{V}^*_{t-1} > V^*_{t-1}$. To recover the known optimal memory $M_t$ we lift $\hat{M}_{t-1}$ to $M_t = f(\hat{M}_{t-1}, \mathcal{X}_{\sqcup})$. This implies $\hat{V}^* > V^*$ which in turn contradicts the purported original optimality of $V^*$ and therefore $\hat{\pi}_E$.*

### 2.2.4  Reward and information collection

We now offer our union of independent curiosity with independent reward collection. We consider a new agent who interacts with an environment and who wishes to maximize both information value *and* reward value, as shown in Eq 5. How can an agent do this? To answer this, let's make two assumptions:

**Assumption 1** (The hypothesis of equal importance). *Reward and information collection are equally important in reinforcement learning.*

**Assumption 2** (The curiosity trick). *Curiosity is a sufficient solution for all exploration problems where learning is possible.*

If reward and information value are equally important then the Bellman answer is to choose which value is larger for any given state and time. If curiosity is sufficient, then we can use this Bellman rule and pursue information value greedily during exploration confident that we will also arrive at a reward collection policy. The info-reward value function $V_{\pi_{ER}}$ for a given policy is defined by a series of local policy maxima,

$$V_{\pi_{ER}}(S) = \max_{\pi_{ER}} \mathbb{E}\left[\sum_{t=0}^{T} \max[E_t, R_t] \;\middle|\; S_t = S\right] \tag{5}$$

Having already established we have a Bellman optimal policy for $E$, and knowing reinforcement learning provides many solutions for $R$ (Sutton & Barto, 2018; Bertsekas, 2017), we can write out the Bellman optimal solution for $V_{\pi_{ER}}$. This is,

$$V_{\pi_{ER}}(S) = \max_{\pi_{ER}} \mathbb{E}\left[\max[E_t, R_t] + V_{\pi_{ER}}(S_{t+1})\middle|\; S_t = S, A_t = A\right]. \tag{6}$$

From here we can substitute in the respective value functions for reward and information value. This gives us Eq.7, which translates to the Bellman optimal decision policy shown in Eq.8.

$$V_{\pi_{ER}}(S) = \max_{\pi_{ER}} \mathbb{E}\left[\max[V_{\pi_E}(S), V_{\pi_R}(S)] + V_{\pi_{ER}}(S_{t+1})\middle|\; S_t = S, A_t = A\right] \tag{7}$$

$$\pi_{ER}(S) = \begin{cases} \pi_R(S) & \text{if } V_{\pi_R}(S) \geq V_{\pi_E}(S) \\ \pi_E(S) & \text{otherwise} \end{cases} \tag{8}$$

### 2.2.5 Simplifying with win-stay lose-switch

In our analysis of decision making under dynamic programming we have assumed that the value for the next state, $V(S_{t+1})$, is readily available with no uncertainty. In practice for most environments this is not the case. This presents a key distinction between the field of dynamic programming and reinforcement learning. If we no longer assume $V_{\pi_E}$ and $V_{\pi_R}$ are available, but must be estimated from the environment, then we are left with something of a paradox for information value.

In order to resolve this paradox, we wish to use value functions to decide between information and reward value but are simultaneously estimating those values. There are a range of methods to handle this (Sutton & Barto, 2018; Bertsekas, 2017), but we opted to further simplify $\pi_{ER}$ in three ways. We first shift from using the full value functions to using only the last payout, $E_{t-1}$ and $R_{t-1}$. Second, we remove all state-dependence leaving only time dependence. Third, we also include $\eta > 0$ to control the duration of exploration. These simplifications give Eq. 9, a win-stay lose-switch (WSLS) rule for exploration-exploitation.

$$\tilde{\pi}_{ER}(S) = \begin{cases} \pi_R(S) & \text{if } R_{t-1} \geq E_{t-1} - \eta \\ \pi_E(S) & \text{otherwise} \end{cases} \tag{9}$$

In Eq. 9, we think of $\pi_R$ and $\pi_E$ as two "players" in a game played for behavioral control (Estes, 1994). We feel the approach has several advantages. It is myopic and therefore can optimally handle nonlinear changes in either the reward function or environmental dynamics (Hocker & Park, 2019). In stationary settings, its regret is bounded (Robbins, 1952). It can approximate Bayesian inference (Bonawitz et al., 2014). It leads to cooperative behavior (Nowak & Sigmund, 1993). Further, WSLS has a long history in psychology, where it predicts and describes human and animal behavior (Worthy & Todd Maddox, 2014). But most of all, our version of WSLS is simple to implement and seems robust in practice (shown below).

## 3 Experiments

### 3.1 Tasks

We evaluated our WSLS exploration policy in seven variants of the bandit task popular in reinforcement learning Sutton & Barto (2018). The general structure of each task was similar. On each trial there were a set of $n$ choices, and the goal of the agent to try and learn the best one, often determined by choosing the target that returns the most rewards, but not always (see below). Each choice action returns a "payout" according to a predetermined probability. Payouts are information, reward, or both (Figure 3).

*Task 1* was designed to examine information foraging. There were no rewards. There were four choices. Three of these generated either a "yellow" or "blue" symbol, with a set probability. See Figure 3**a**.

*Task 2* was a simple bandit, designed to examine reward collection. At no point does the task generate information. Rewards were 0 or 1. There were four choices. The best choice had a payout of $p(R = 1) = 0.8$. This is a much higher average payout than the others ($p(R = 1) = 0.2$). See Figure 3**b**.

*Task 3* was designed with very sparse rewards (Silver et al., 2016; 2018). There were 10 choices. The best choice had a payout of $p(R = 1) = 0.02$. The other nine had, $p(R = 1) = 0.01$ for all others. See Figure 3c.

*Task 4* had deceptive rewards. By deceptive we mean that the best long-term option presents itself initially with a lower value. The best choice had a payout of $p(R > 0) = 0.6$. The others had $p(R > 0) = 0.4$. Value for the best arm dips, then recovers. This is the "deception" It happens over the first 20 trials. Rewards were real numbers, between 0-1. See Figure 3d.

Figure 3: Multi-armed bandits – illustration and payouts. On each trial the agent must take on $n$ actions. Each action generates a payout. Payouts can be information, reward, or both. For comments on general task design. **a.** A 4 choice bandit for information collection. In this task the payout is information, a yellow or blue "stimulus". A good agent should visit each arm, but quickly discover that only arm two is information bearing. **b.** A 4 choice design for reward collection. The agent is presented with four actions and it must discover which choice yields the highest average reward. In this task that is Choice 2. **c.** A 10 choice sparse reward task. Note the very low overall rate of rewards. Solving this task with consistency means consistent exploration. **d.** A 10 choice deceptive reward task. The agent is presented with 10 choices but the action which is the best in the long-term (>30 trials) has a lower value in the short term. This value first declines, then rises (see column 2). **e.** A 121 choice task with a complex payout structure. This task is thought to be at the limit of human performance. A good agent will eventually discover choice number 57 has the highest payout. **f.** This task is identical to *a.*, except for the high payout choice being changed to be the lowest possible payout. This task tests how well different exploration strategies adjust to simple but sudden change in the environment.

*Tasks 5-6* were designed with 121 choices, and a complex payout structure. Tasks of this size are at the limit of human performance (Wu et al., 2018). We first trained all agents on *Task 6*, whose payout can be seen in Figure3f-g. This task, like the others, had a single best payout $p(R = 1) = 0.8$. After training for this was complete, final scores were recorded as reported, and the agents were then challenged by *Task 7*. *Task 7* was identical except that the best option was changed to be the worst $p(R = 0) = 0.2$ (Figure 3f-g).

### 3.2 Agents

We considered two kinds of specialized agents to play our set of tasks. Those suited to solve bandit tasks and those suited to solve foraging tasks.

*Bandit: E-explore* - This agent uses our algorithm defined above. It pursued either pure exploration based on $E$ maximization or pure exploitation based on pure $R$ maximization. All its actions are deterministic and greedy. Both $E$ and $R$ maximization was implemented as an actor-critic architecture (Sutton & Barto, 2018) where value updates were made in crtic according to Eq. 18. Actor action selection was governed by,

$$A_t = \arg\max A \in \mathbb{A} = Q(., A_t) \tag{10}$$

where we use the "." to denote the fact our bandits have no meaningful state.

This agent has two parameters, the learning rate $\alpha$ (Eq. 18) and the boredom threshold $\eta$ (Eq. 9). Its payout function was simply, $G = R_t$ (Eq. 18).

*Bandit: Reward.* An algorithm whose objective was to estimate the reward value of each action, and stochastically select the most valuable action in an effort to maximize total reward collected. It's payout function was simply, $G = R_t$ (Eq. 18). Similar to the E-explore agent, this agent used actor-critic, but its actions were sampled from a softmax / Boltzman distribution,

$$p(A_t) = \frac{e^{\gamma Q_R(.,A_t)}}{\sum_{A \in \mathbb{A}} e^{\gamma Q_R(.,A)}} \tag{11}$$

Where $\gamma$ is the "temperature" parameter. Large $\gamma$ generate more random actions. This agent has two parameters, the learning rate $\alpha$ (Eq. 18) and the temperature $\gamma > 0$ (Eq. 11).

*Bandit: Reward+Info.* This algorithm was based on the Reward algorithm defined previously but it's reward value was augmented or mixed with $E$. Specifically, the information gain formulation of $E$ described below. Its payout is, $G_t = R_t + \beta E_t$ (Eq. 18). Critic values were updated using this $G$, and actions were selected according to Eq. 11. This agent has three parameters, the learning rate $\alpha$ (Eq. 18) the temperature $\gamma$ (Eq. 11) and the exploitation weight $\beta > 0$. Larger values of $\beta$ will tend to favor exploration.

*Bandit: Reward+Novelty.* This algorithm was based on the Reward algorithm defined previously but its reward value was augmented or mixed with a novelty/exploration bonus (Eq. 12).

$$B_{A_t} = \begin{cases} B & \text{if } A_t \text{ not it } \mathbb{Z} \\ 0 & \text{otherwise} \end{cases} \tag{12}$$

Where $\mathbb{Z}$ is the set of all actions that agent has taken so far in a simulation. Once all actions have been tried, $B = 0$.

This agent's payout is, $G_t = R_t + B_t$ (Eq. 18). Critic values were updated using this $G$ and actions were selected according to Eq. 11. This agent has three parameters, the learning rate $\alpha$ (Eq. 18), the temperature $\gamma$ (Eq. 11), and the bonus size $B > 0$.

*Bandit: Reward+Entropy.* This algorithm was based on the Reward algorithm defined previously but its reward value was augmented or mixed with an entropy bonus (Eq. 13). This bonus was inspired by the "softactor" method common to current agents in the related field of deep reinforcement learning (Haarnoja et al., 2018).

$$H(A_t) = \sum_{A \in \mathbb{A}} p(A) \log p(A) \tag{13}$$

Table 1: Exploration strategies for bandit agents.

| Name | Class | Exploration strategy |
|---|---|---|
| E-explore | Info. val. | Deterministic max. of information value |
| Random | Random | Random exploration |
| Reward | Mixed | Softmax sampling of reward |
| Reward+Info. | Mixed | Random sampling of reward $+ \beta$ information value |
| Reward+Novelty | Mixed | Random sampling of reward $+ \beta$ novelty signal |
| Reward+Entropy | Mixed | Random sampling of reward $+ \beta$ action entropy |
| Reward+EB | Mixed | Random sampling of reward $+ \beta$ visit counts |
| Reward+UCB | Mixed | Random sampling of reward $+ \beta$ visit counts |

Where $p(A)$ was estimated by a simple normalized action count.

This agent's payout is, $G_t = R_t + \beta H_t$ (Eq. 18). Critic values were updated using this $G$, and actions were selected according to Eq. 11. It has three free parameters, the learning rate $\alpha$ (Eq. 18), the temperature $\gamma$ (Eq. 11), and the exploitation weight $\beta > 0$.

*Bandit: Reward+EB.* This algorithm was based on the Reward algorithm defined previously but it's reward value was augmented or mixed with an evidence bound (EB) statistic (Bellemare et al., 2016) (Eq 14).

$$\text{EB}(A) = \sqrt{C(A)} \tag{14}$$

Where $C(A)$ is a running count of each action, $A \in \mathbb{A}$. This agent's payout is, $G_t = R_t + \beta \text{ EB}$ (Eq. 18). Critic values were updated using this $G$, and actions were selected according to Eq. 15.

$$A_t = \arg\max A \in \mathbb{A} = Q(., A_t) + \beta \text{ EB}(A) \tag{15}$$

It has two parameters, the learning rate $\alpha$ (Eq. 18) and the exploitation weight $\beta > 0$.

*Bandit: Reward+UCB.* This algorithm was based on the Reward algorithm defined previously, but it's reward value was augmented or mixed with an upper confidence bound (UCB) statistic (Bellemare et al., 2016) (Eq 16).

$$\text{UCB}(A) = \frac{2 \log(N_A + 1)}{\sqrt{C(A)}} \tag{16}$$

Where $C(A)$ is a running count of each action, $A \in \mathbb{A}$ and $N_A$ is the total count of all actions taken. This agent's payout is, $G_t = R_t + \beta \text{ UCB}$ (Eq. 18). Critic values were updated using this $G$, and actions were selected according to Eq. 17.

$$A_t = \arg\max A \in \mathbb{A} = Q(., A_t) + \beta \text{ UCB}(A) \tag{17}$$

It has three two parameters, the learning rate $\alpha$ (Eq. 18) and the exploitation weight $\beta > 0$.

In the majority of simulations we have used a Bayesian or Information Gain (IG) formulation for curiosity and $E$. Each task's observations fit a simple discrete probabilistic model, with a memory "slot" for each action for each state. Specifically, probabilities were tabulated on state reward tuples, $(S, R)$. To measure distances in this memory space we used the Kullback–Leibler divergence (Goodfellow et al., 2016; Itti & Baldi, 2009; López-Fidalgo et al., 2007; Schmidhuber, 2008; Ganguli & Sompolinsky, 2010; Ay, 2015).

Reward and information value learning for all agents on the bandit tasks were made using update rule below,

$$V(S) = V(S) + \alpha[G_t - V(S_t)] \tag{18}$$

Table 2: Hyperparameter tuning - parameters and ranges.

| Task | Agent | Parameter | Range |
|---|---|---|---|
| Bandit | E-explore | $\eta$ | (1e-9, 1e-2) |
| Bandit | E-explore | $\alpha$ | (0.001, 0.5) |
| Forage | E-explore | $\eta$ | (1e-9, 1e-2) |
| Forage | E-explore | $\alpha$ | (0.001, 0.5) |
| Bandit | Reward | $\gamma$ | (0.001, 1000) |
| Bandit | Reward | $\alpha$ | (0.001, 0.5) |
| Bandit | Reward+Info. | $\gamma$ | (0.001, 1000) |
| Bandit | Reward+Info. | $\beta$ | (0.001, 10) |
| Bandit | Reward+Info. | $\alpha$ | (0.001, 0.5) |
| Forage | Reward+Info. | $\gamma$ | (0.001, 1000) |
| Forage | Reward+Info. | $\beta$ | (0.001, 10) |
| Forage | Reward+Info. | $\alpha$ | (0.001, 0.5) |
| Bandit | Reward+Novelty | $\gamma$ | (0.001, 1000) |
| Bandit | Reward+Novelty | $B$ | (1, 100) |
| Bandit | Reward+Novelty | $\alpha$ | (0.001, 0.5) |
| Bandit | Reward+Entropy | $\gamma$ | (0.001, 1000) |
| Bandit | Reward+Entropy | $\beta$ | (0.001, 10) |
| Bandit | Reward+Entropy | $\alpha$ | (0.001, 0.5) |
| Bandit | Reward+EB | $\beta$ | (0.001, 10) |
| Bandit | Reward+EB | $\alpha$ | (0.001, 0.5) |
| Bandit | Reward+UCB | $\beta$ | (0.001, 10) |
| Bandit | Reward+UCB | $\alpha$ | (0.001, 0.5) |

Where $V(S)$ is the value for each state, $G_t$ is the *return* for the current trial, either $R_t$ or $E_t$, and $\alpha$ is the learning rate $(0 - 1]$. See the *Hyperparameter optimization* section for information on how $\alpha$ is chosen for each agent and task.

Value learning updates for all relevant agents in the foraging task were made using the TD(0) learning rule (Sutton & Barto, 2018),

$$V(S_t) = V(S_t) + \alpha[G_t - V(S_{t+1}) - V(S_t)] \tag{19}$$

We assume an agent will have a uniform prior over the possible actions $\mathbb{A}$ and set accordingly, $E_0 = \sum_K p(A_k) \log p(A_k)$.

The hyperparameters for each agent were tuned independently for each task by random search (Bergstra & Bengio, 2012). Generally, we reported results for the top 10 values, sampled from 1000 possibilities. Top 10 parameters and search ranges for all agents are shown in Supplemental Table 2. Parameter ranges were held fixed for each agent across all the bandit tasks.

### 3.3 Information collection simulations

The optimal policy to maximize information value is not a sampling policy. It is a deterministic greedy maximization (Eq 3). In other words, to minimize uncertainty during exploration an agent should not introduce additional uncertainty by taking random actions (Sehnke et al., 2010). To confirm our deterministic method is best we examined a multi-armed information task in Task 1 (Figure 3a). This variation of a bandit task replaced rewards with state information that must be intrinsically valued. In this case, state is simple colors represented internally as integers. On each selection, the agent saw one of two colors (integers) returned according to specific probabilities (Figure 3a).

Figure 4 shows the results of this head-to-head comparison of information-maximizing agents. None of the reward collection agents were run on this task because no rewards are provided, just feedback state, making

optimization impossible for these agents. As expected, the deterministic information-maximizing agent was able to converge to its best estimate of the more informative target faster than the stochastic agent (Figure 4a-b), even though both agents used an otherwise identical curiosity-directed search. In addition, the deterministic agent was able to accrue a higher estimate of $E$ than the stochastic agent (Figure 4c), with fewer exploratory steps (Figure 4d). This confirms the utility of our deterministic search policy.

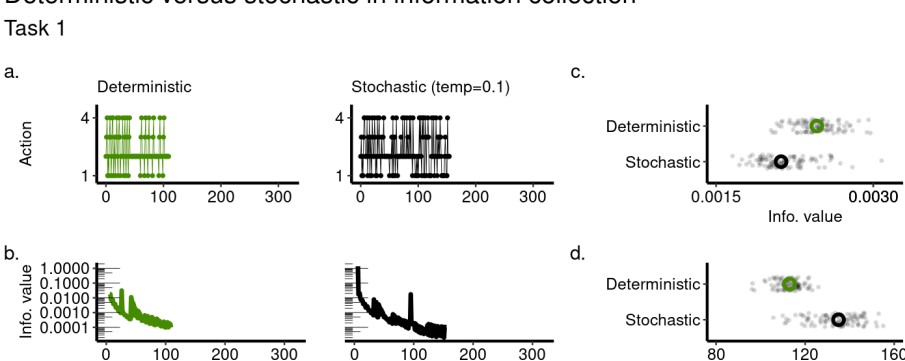

Figure 4: Comparing deterministic versus stochastic variations of the same curiosity algorithm in Task 1. Deterministic results are shown in the left column and stochastic results are shown in the right. **a**. Examples of target selection, across trials, for deterministic (green) and stochastic (black) information-seeking agents. The block ended when $E$ converged to a minimum value. **b**. Information value, $E$, plotted across time. Matches the behavior shown in a. **c**. Average information value for 100 independent simulations. er of steps it took to reach the stopping criterion in c., e.g. the boredom threshold *eta* (described below). Smaller numbers of steps imply a faster search. **d**. Number of exploration steps until convergence for each agent.

## 3.4 Reward collection simulations

Now that we know that our agent can maximize information, we can look at how our *E-explore* agent compares with more traditional reward maximization agents in typical bandit tasks. Our overall goal is to answer the question: can curious search solve reward collection problems better than other approaches which use reward value? To find out we measured the total reward collected in several bandit tasks (Figure 3b-g). We considered eight agents, including ours (Table 1), described in Section 3.2.

Figure 5 presents the overall performance in some standard bandit tasks: simple (Figure 5b), sparse (Figure 5c), high dimensional (Figure 5e) and non-stationary (Figure 5f). Overall, our approach matches or sometimes exceeds all the other exploration strategies (Figure 5a)). Despite our *E*-exploration agent never optimizing for reward value, our method of pure exploration (Bubeck et al., 2010) matched, or in some cases outperformed standard approaches that rely on reward value. Indeed, with the exception of Task 4, performance improvements, when present, were small but meaningful. This modest improvement in most tasks might not look noteworthy. We argue it is because our exploration strategy never optimizes for reward value explicitly. Yet, we meet or exceed exploration strategies which do. In other words, when intrinsic curiosity is used in an optimal tradeoff with extrinsic reward, curiosity does seem sufficient in practice.

However, our information maximizing strategy was highly effective in one particular context that is typically challenging for reward maximizing agents: deceptive feedback. Task 4, the deception task (Figure 3d), involved an initial, misleading, 20 step decline in reward value for the optimal target. Comparing the *E*-exploration agent against the reward maximizing agents (Figure 5d), we see that the other exploration strategies produced little better than chance performance. When deception is present, externally motivated exploration is a liability. Yet *E*-explore was highly effective in this context because the change in feedback structure over time creates a boost in $E$, pushing the agent to seek information on the changing target and becoming robust to this early deception paradigm.

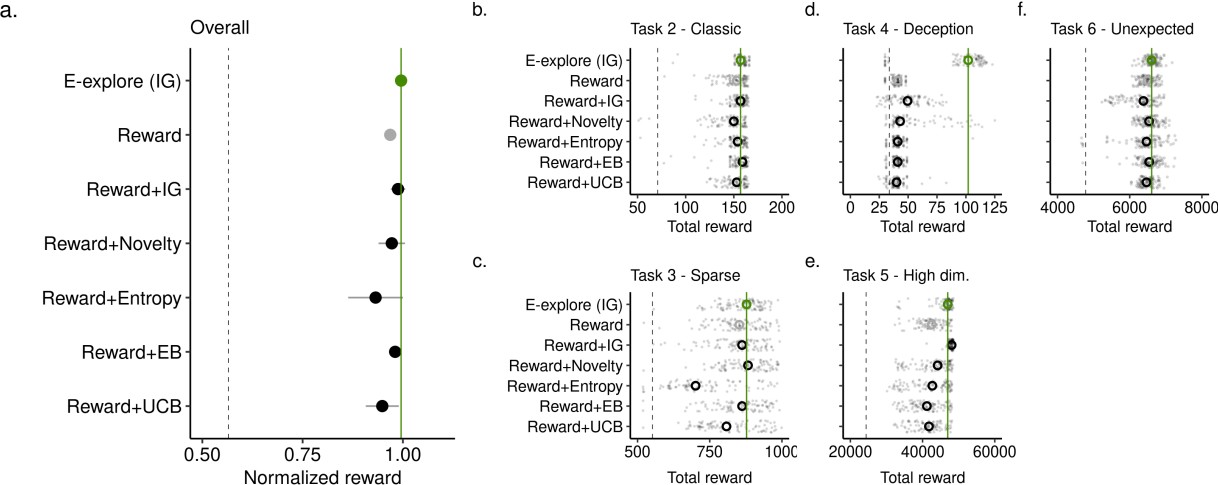

Figure 5: Reward collection performance. **b.** Overall results. Total reward collected for each of the four tasks was normalized. Dot represents the median value, error bars represent the median absolute deviation between tasks (MAD). **b.** Results for Task 2, which has four choices and one clear best choice. **c.** Results for Task 3, which has 10 choices and very sparse positive returns. **d.** Results for Task 4, whose best choice is initially "deceptive" in that it returns suboptimal reward value over the first 20 trials. **e.** Results for Task 6, which has 121 choices and a quite heterogeneous set of payouts but still with one best choice. **f.** Results for Task 7, which is identical to Task 6 except the best choice was changed to the worst. Agents were pre-trained on Task 6, then tested on 7. In panels b-e, grey dots represent total reward collected for individual experiments while the large circles represent the experimental median.

Finally, another place where we see a clear boost in performance with *E*-explore is in high-dimensional action spaces (Task 5). This task, modeled against the upper limit of human action selection (Wu et al., 2018), forced agents to explore across 121 potential discrete actions to find the optimal choice (Figure 3f). While we see that our *E*-explore agent performed as well as a standard reward maximization agent with a novelty boost (Figure 5e), our agent that maximizes *E* does so with greater efficiency (Figure 6a), converging on the optimal target faster than any other agent, and settling on a stable reward return by the end of each simulation run (Figure 6b). Of course, once the optimal target in this high-dimensional space becomes the *least* optimal target (Task 7), the *E*-exploration agent converges on the new optimal target, with a slightly higher reward return than the other agents (Figure 5f).

Of course, one could argue that the performance of our *E*-explore agent reflects the specific hyperparameter choices used to initialize the models. Thus we tested the robustness of all the agents to environmental mistunings (Figure 7). We re-examined total rewards collected across all model tunings or hyperparameters. Here the *E*-explore agent was markedly better than the other agents, with both substantially higher total reward when we integrate over all parameters (Figure 7a) and when we consider the tasks individually (Figure 7b-f).

## 4 Discussion

Here we presented a new way to think about solutions to the classic exploration-exploitation problem. Using theory and simulations we argue that reinforcement learning practitioners can safely set aside any intuitions that suggest a purely curious search is too inefficient to be generally practical when the goal is reward collection. To begin with, we derived a measure of information value that is possible to directly measure in most any memory system, especially with gradient based learning. This measure also properly

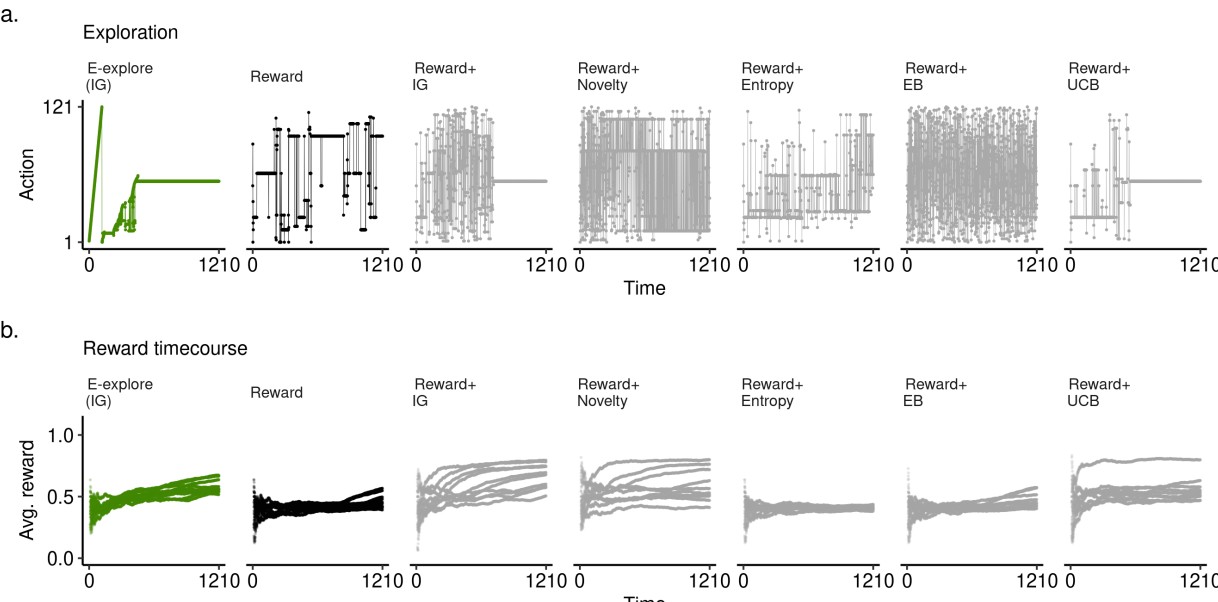

Figure 6: Behavior on a complex high-dimensional action space (Task 5). **a**. Examples of strategies for all agents (1 example). **b**. Average reward time courses for all agents (10 examples).

generalizes many prior efforts to formalize curiosity in computer science, biology, and psychology. Using this measure of information value, $E$, we then showed how it can be maximized using standard reinforcement learning algorithms. This dynamic competition between pure curiosity and pure reward collection performs well on a variety of explore-exploit problems. Curiosity-driven exploration, tempered by boredom, matches or exceeds reward-based strategies when rewards are dense, sparse, high-dimensional, or non-stationary. In addition, curiosity driven exploration uniquely overcomes deceptive feedback schedules that can trip up reward maximizing agents. Our game between curiosity and rewards also seems far more robust than standard algorithms to hyperparameter choices.

While the theoretical and empirical results presented here make a strong case for thinking of maximizing information versus maximizing reward as a scheduling problem, depending on relative immediate value, a lot of questions remain unanswered. Here we address some of the more immediate questions about our approach.

### 4.1 Questions and answers

**Why curiosity?** Not all situations involve immediately maximizing tangible rewards. Many natural (and artificial) behaviors involve exploring just to know. Curiosity as an algorithm is highly effective at solving difficult optimization problems (Schmidhuber, 1991; Pathak et al., 2017; Stanton & Clune, 2018; Fister et al., 2019; Mouret & Clune, 2015; Colas et al., 2020; Cully et al., 2015; Pathak et al., 2017; Schwartenbeck et al., 2019; Laversanne-Finot et al., 2018).

**Is this explore-exploit game a slight of hand theoretically?** Yes. We have taken one problem, the explore-exploit tradeoff, that is formally considered intractable and replaced it with another problem that can be solved. In this replacement we swap one behavior, exploration with reward seeking in mind, for another, curiosity (i.e., exploration to gather information).

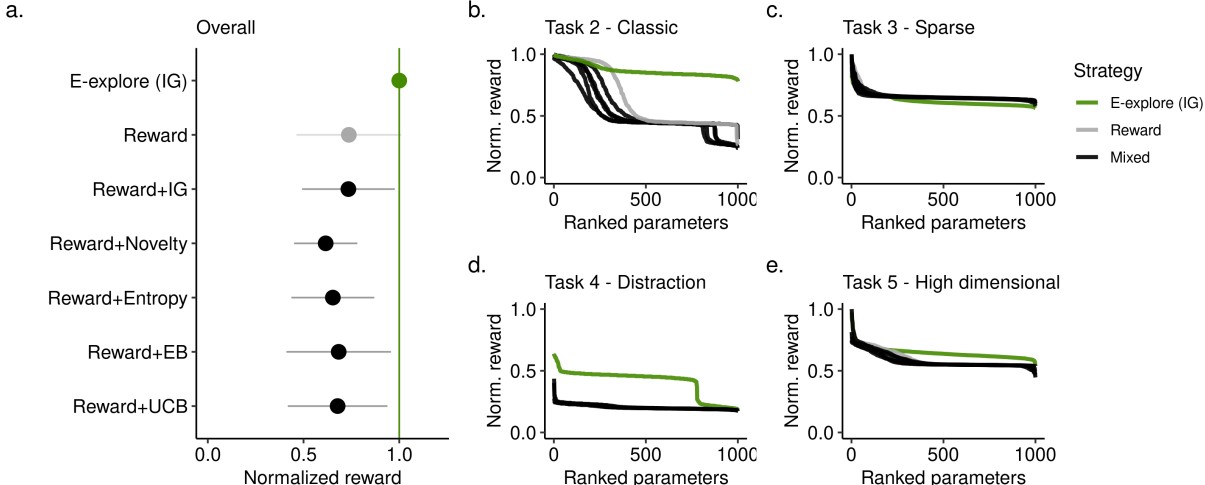

Figure 7: Exploration hyperparameter sensitivity. **a**. Integrated total reward (normalized) across 1000 randomly chosen exploration hyperparameters. Dots represent the median. Error bars are the median absolute deviation. **b-f**. Normalized total reward for exploration 1000 randomly chosen parameters, ranked according to performance. Each subpanel shows performance on a different task. Lines are colored according to overall exploration strategy - *E*-explore, reward only, or a mixed value approach blending reward and an exploration bonus).

**Is this too complex?** Perhaps turning a single objective into two, as we have, is too complex an answer for reinforcement learning. If this is true, then it would mean our strategy is not parsimonious. Should we reject it on that alone?

Questions about parsimony can be sometimes resolved by considering the benefits versus the costs of added complexity. The benefits are an optimal value solution to the exploration-exploitation tradeoff. A solution that seems especially robust to model-environment mismatch. At the same time, curiosity-as-exploration can build a model of the environment (Ha & Schmidhuber, 2018), useful for later planning (Ahilan et al., 2019; Poucet, 1993), creativity, imagination (Schmidhuber, 2010), while also building diverse action strategies (Lehman & Stanley, 2011b; Lehman et al., 2013; Mouret & Clune, 2015; Colas et al., 2020).

**Is this too simple?** Our game between curiosity and reward seems to be simple to solve. This is suspicious given how much work has been done on explore-exploit problems in the past. So have we "cheated" by changing the problem in the way we have?

The truth is that we might have cheated in one sense: the tradeoff might really have to be as hard as it has seemed in the past. But the general case for curiosity is clear and backed up by the brute fact of its widespread presence in the animal kingdom Byrne (2013). The real questions are: 1) is curiosity so useful and so robust that it is sufficient for all exploration with learning (Fister et al., 2019), and 2) is curiosity really equally important as tangible rewards?

The answer to these questions is, we believe, empirical. If our account does well in describing and predicting animal behavior, that would be some evidence for it (see (Sumner et al., 2019; Wang & Hayden, 2019; Jaegle et al., 2019; Gottlieb & Oudeyer, 2018; Kidd & Hayden, 2015; Berlyne, 1950; Colas et al., 2020; Rahnev & Denison, 2018; Wilson et al., 2020; Berger-Tal et al., 2014)). If it predicts neural structures related to exploratory behaviors (see (Cisek, 2019; Kobayashi & Hsu, 2019)), then that would be other evidence for our theory. If our theory proves useful in machine learning applications (see (Burda et al., 2018; Schmidhuber, 1991; de Abril & Kanai, 2018; Fister et al., 2019; Lehman & Stanley, 2011b; Stanley & Miikkulainen, 2004; Colas et al., 2020; Cully et al., 2015; Wilson et al., 2020; Pathak et al., 2019)) then that would be be even

more evidence for it. In other words how simple, complex, or parsimonious a theoretical idea is comes down to its usefulness. That is for follow up work to decide.

**What about truth?** In other prototypical approaches, information value comes from prediction errors or is otherwise measured by how well learning corresponds to the environment (Behrens et al., 2007; Kolchinsky & Wolpert, 2018; Tishby et al., 2000) or how useful the information might be in the future (Dubey & Griffiths, 2020). Colloquially, one might call this "truth seeking". As a people we pursue information that is fictional (**?**), or based on analogy (**?**), or outright wrong (**?**). It is our view that conspiracy theories, disinformation, and more mundane errors, are far too commonplace for information value to rest on mutual information or error alone. This does not mean that holding false beliefs cannot harm survival, but this harm might be a second order question as far as information value goes.

**What about information theory?** The short answer is that the problems of communicating information and the value of that information, are wholly different issues that require different theories. Shannon and Weaver (Shannon & Weaver, 1964) in their classic introduction to the topic, describe information as a communication problem with three levels. **A.** The technical problem of transmitting accurately. **B.** The semantic problem of meaning. **C.** The effectiveness problem of using information to change behavior. They then go on to describe how communication theory addresses only problem **A.**, which has lead to information theory finding a broad application over the last 75 years.

Yet, valuing information is not, at its core, a communications problem. It is a problem in personal history. Consider Bob and Alice who are having a phone conversation, and then Tom and Bob who have the exact same phone conversation. The personal history of Bob and Alice will determine what Bob learns in their phone call, and so determines what he values in that phone call, or so we argue. This might be different from what Bob learns when talking to Tom, even when the conversations were identical. What we mean to show by this example is that the personal history, also known as past memory, defines what is valuable on a channel.

We have summarized this diagrammatically, in Figure 8.

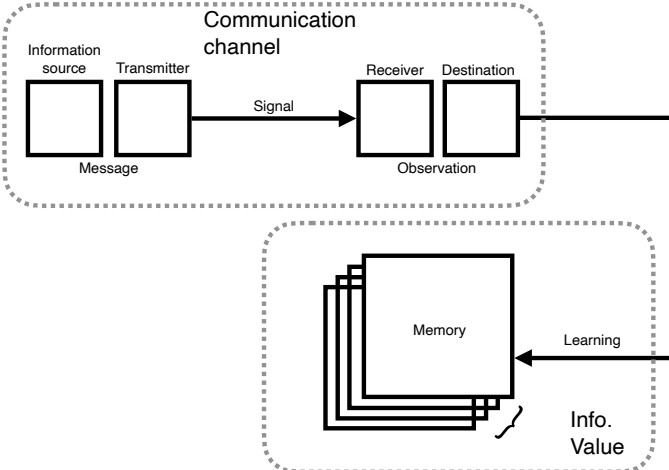

Figure 8: The relationship between the technical problem of communication with the technical problem of value. Note how value is not derived from the channel directly. Value comes from learning about observations taken from the channel, which in turn depends on memory.

Finally, we argue there is an analogous set of levels for information value as those that Shannon and Weaver describe for information. There is the technical problem of judging how much was learned. This is the one we address. There is the semantic problem of what this learning "means" (see Dretske (1981)) and also what

its consequences are. There is also the effectiveness problem of using what was learned to some other effect. These are applications of the theoretical approach that we propose here.

**Does value as a technical problem even make sense?** Having a technical definition of information value, free from any meaning or external references beyond memory dynamics, might seem counterintuitive for any value measure. We suggest that it is not any more or less counterintuitive than stripping information of meaning.

**So you suppose there is always a positive value for learning of all fictions, disinformation, and deceptions?** According to our framework, we must. This is a most unexpected prediction. Yet it seems consistent with the behavior of humans, and animals alike. Humans do consistently seek to learn falsehoods. Our axioms suggest why this is rational: it is good learning progress from the perspective of changes in memory.

**Was it necessary to build a general theory for information value to describe curiosity?** We made an idealistic choice that worked out. The field of curiosity studies has shown that there are many kinds of curiosity (Kidd & Hayden, 2015; Gottlieb & Oudeyer, 2018; Berlyne, 1950; Lehman & Stanley, 2011b; Pathak et al., 2017; Schmidhuber, 1991; 2008; Schwartenbeck et al., 2019; de Abril & Kanai, 2018; Stanton & Clune, 2018; Zhou et al., 2020; Loewenstein, 1994; Kashdan et al., 2019; Keller et al., 1994; Wang & Hayden, 2020). At the extreme limit of this diversity is a notion of curiosity defined for any kind of observation, and any kind of learning. This is what we offer. At this limit we can be sure to span fields, addressing the problem of information value as it appears in computer science, machine learning, game theory, psychology, neuroscience, biology, economics, among others.

**What about other models of curiosity?** Curiosity has found many specific definitions over the years (Berlyne, 1950; Oudeyer, 2018; Gottlieb & Oudeyer, 2018). Curiosity has been described as a prediction error that drives learning progress (Schmidhuber, 1991; Kaplan & Oudeyer, 2007). Schmidhuber (Schmidhuber, 1991) noted the advantage of looking at the derivative of errors, rather than errors directly, which falls within our definition of information value. Itti (Itti & Baldi, 2009) and others (Schmidhuber, 2008; Friston et al., 2017; Reddy et al., 2016; Calhoun et al., 2015) have taken a statistical and Bayesian view often using the KL divergence to estimate information gain or Bayesian surprise. This also falls under our general definition of information value. Other approaches have been based on adversarial learning, model disagreement (Pathak et al., 2017), or model compression (Schmidhuber, 2008). Some measures focused on past experience as a driver to seek information (Kim et al., 2020; Schmidhuber, 1991; Ha & Schmidhuber, 2018) while others focused on future planning and imagination (Savinov et al., 2019; Colas et al., 2020; Mendonca et al., 2021). These are, again, consistent within our general defition of information value. Finally, graphical models of memory have been useful in explaining how curiosity drives understanding (Colas et al., 2020), which are similar in farming to our geometric model.

What distinguishes our approach to defining information value is that we focus on memory dynamics, and base value on some general axioms that braodly cover these prior definitions. We try to embody the idea of curiosity "as learning for learning's sake", not done for the sake of some particular goal (Gabaix et al., 2006), or for future utility (Dubey & Griffiths, 2020). Our axioms were however designed with all these other definitions in mind. In other words, we do not aim to add a new metric. We aim to generalize others.

**Is information a reward?** If reward is any quantity that motivates behavior, then our definition of information value is a reward, an intrinsic reward. This last point does not mean that information value and environmental rewards are interchangeable however. Rewards from the environment are a conserved resource, information is not. For example, if a rat shares a potato chip with a cage-mate, it must break the chip up leaving it less food for itself. While if a student shares the Pythagorean theorem with a classmate, that student does not have less knowledge of the Pythagorean theorem as a result.

**But isn't curiosity impractical for many problems?**

Let's consider the answer to this by discussing colloquially how science and engineering often interact. Science is seen as an open-ended inquiry, whose goal is knowledge but whose practice is driven by learning progress. This is a fairly impractical aim in terms of finding solutions to problems. Engineering, on the other hand, is often seen as finding specific solutions to specific problems. This is both fairly practical, but also limited

by the existing base of scientific knowledge. They each have their own pursuits, in other words, but they also learn from each other often in alternating iterations. Their different objectives is what makes them such good long-term collaborators.

In a related view Gupta et al (Gupta et al., 2006) encouraged managers in business organizations to strive for a balance in the exploitation of existing markets and ideas with exploration. They suggested managers should pursue periods of "punctuated equilibrium", where employees work towards either pure market exploitation or pure curiosity-driven exploration to drive future innovation.

**Is the algorithmic run time for curiosity practical?** According to our model, there is a lower limit in algorithmic run time for curiosity. The worst case algorithmic run time of our meta-policy method is linear and additive in its independent policies. If it takes $T_E$ steps for $\pi_E$ to converge, and $T_R$ steps for $\pi_R$, then the worst case run time for $\pi_{ER}$ is $T_E + T_R$. This puts our model at the same run time efficiency as standard reinforcement learning models.

**Does this mean you are hypothesizing that boredom is actively tuned?** Yes we are predicting that. In fact, Geana and Daw (Geana & Daw, 2016) showed preciselyl this in a series of experiments. They reported that altering the expectations of future learning in turn alters self-reports of boredom. Others have shown how boredom and self-control interact to drive exploration (Hill & Perkins, 1985; Bench & Lench, 2013; Wolff & Martarelli, 2020). So boredom does seem to be a pliable, and likely actively tuned, parameter.

**Do you have any evidence of your theory in animal behavior or neural circuits?** There is some evidence for our theory of curiosity in psychology and neuroscience, however, in these fields curiosity and reinforcement learning have largely developed as separate disciplines (Berlyne, 1950; Kidd & Hayden, 2015; Sutton & Barto, 2018). Indeed, we have highlighted how they are separate problems, with links to different basic needs: gathering resources to maintain physiological homeostasis (Keramati & Gutkin, 2014; Juechems & Summerfield, 2019) and gathering information to decide what to learn and to plan for the future (Valiant, 1984; Sutton & Barto, 2018). Here we suggest that, though they are separate problems, they are problems that can generally solve one another. This insight is the central idea to our view of the explore-exploit decisions.

Yet there are hints of this independent cooperation of curiosity and reinforcement learning out there. Cisek (2019) has traced the evolution of perception, cognition, and action circuits from the Metazoan to the modern age (Cisek, 2019). The circuits for reward exploitation and observation-driven exploration appear to have evolved separately, and act competitively, exactly the model we suggest. In particular he notes that exploration circuits in early animals were closely tied to the primary sense organs (i.e. information) and had no input from the homeostatic circuits (Keramati & Gutkin, 2014; Cisek, 2019; Juechems & Summerfield, 2019). This neural separation for independent circuits has been observed in some animals, including zebrafish (Marques et al., 2019) and monkeys (White et al., 2019; Wang & Hayden, 2019).

**Acknowledgments**

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
