# OpenReview forum: "Making a game out of exploration-exploitation"
_TMLR — Rejected by TMLR_

### Review · Reviewer_Crq3 · 2023-05-29

**Summary Of Contributions:**

This paper proposes to view the classic exploration exploitation problem as a game between two agents, one trying to maximize an intrinsic curiosity reward, the other trying the maximize the environment reward. The paper then defines an approach to define information value in an environment that can be applied to exisiting exploration approaches. They show that this quantity follows a Bellman equation and lead to another Bellman equation for the exploration exploitation problem where the agent must choose at every timestep between the exploration and exploitation policies. They then simplify this equation and derive a strategy they call "win-stay lose-switch". They evaluate this strategy against several existing exploration strategies and show this approach is competitive in simple bandits environments.

**Audience:**

Yes

**Broader Impact Concerns:**

No broader impact concerns

**Claims And Evidence:**

No

**Requested Changes:**

List of changes I would like to see:
- Highlight more the relationship between sections 2.2 to 2.3.1 and the rest of manuscript.
- Provide more explanations why equation (14)-(15) is not / could not be directly solved and the WSLS strategy is considered instead.
- Can conjecture 1 be proven instead of being an hypothesis?


Small typos:

- Section 2.3.1 "at the scales we concerned with"
- After equation (24) "it has three two parameters"
- Bandit: Reward + Novelty "its reward value wass augmented"


**Strengths And Weaknesses:**

**Strengths**:

- Overall I found that this is an interesting paper and an interesting approach to the classic exploration exploitation problem.
- I like that the authors attempt to present a unified approach for existing exploration methods.
- The literature review is quite extensive.
- I found the question and answers section before the conclusion helpful.

**Weaknesses**:

- I found the paper difficult to follow, sections 2.2 to 2.3.1 were quite disconnected from the rest of the paper and I am not too sure I understand how they fit with the remaining of the manuscript. For instance an example of memory structure is never provided after section 2.3.1
- I am also sure I understand about the value of work done to derive Equations (12) to (15) when this assumption is then simplified using the WSLS strategy and the main difficult problem is not considered later on.
- Can you provide more details about the validity of Conjecture 1 and 2 in section 2.3.2? Immediate reward and information collection are equally important only if information collection leads to higher reward in the long run? Do you think you show that?
- I don't think I understand the derivation of Equation (15) for Equation (14), could you provide more details?
- How does this framework fit with the existing literature on exploration that considers regret and PAC-MDP results? A discussion on that topic would have been appreciated as the manuscript attempts to unify many existing methods. For example can Equation (22) be rewritten in the form of the Equation (13).
- In the experiment section why only consider bandits problems when the paper is about exploration exploitation in reinforcement learning?
- Figures 4 and 5 are really small and hard to read.

---

> ### Author Response · Authors · 2023-07-05
>
> We thank the reviewer for their helpful suggestions. A detailed response follows.
>
> >  I found the paper difficult to follow, sections 2.2 to 2.3.1 were quite disconnected from the rest of the paper and I am not too sure I understand how they fit with the remainder of the manuscript. For instance an example of memory structure is never provided after section 2.3.1 [x]
>
> On review we also found these connections to be lacking, and we thank the reviewer for their help. We added the following text to the start of 2.2, hoping to motivate this section and 2.3.
>
> "Our aim is to integrate independent value functions for reward collection and information collection into a theoretical whole. This means for us two things, we need to derive a Bellman equation for information collection inline with those equations used for reward collection. We must also feel that we feel introduce a notion of information value that is as general as possible. The former happens in this section. The latter we do axiomatically in the next section. In using axioms we are hoping to build a notion of information value that is nearly as general as information theory."
>
> > I am also sure I understand about the value of work done to derive Equations (12) to (15) when this assumption is then simplified using the WSLS strategy and the main difficult problem is not considered later on.
>
> > Can you provide more details about the validity of Conjecture 1 and 2 in section 2.3.2? Immediate reward and information collection are equally important only if information collection leads to higher reward in the long run? Do you think you show that?
>
> We do not at all suppose that information collection requires direct improvements in reward collection. We have tried to make this clear in the new introduction,
>
> "Our focus on curiosity is in very much the same spirit. Animals in the natural world often strongly prefer to satisfy curiosity over receiving tangible food or water rewards from their environment (food, water, etc). They do so even when that information is costly in terms of time, energy, or both \citep{Song2019,Wang2019,Taylor1975,Singh1970}. This fact stands in contrast to the common assumption in reinforcement learning that environmental rewards are the primary aim. A reader accustomed to thinking of animals as externally motivated by rewards might wonder why animals should show such curious preferences? The answer is simple. Curiosity is a profoundly useful strategy which appears universally across the animal kingdom, as reviewed in \cite{Loewenstein1994, Kidd2015}. As reviewed below, this exploration strategy does not appear directly linked to reward collection, but is instead essential for agent survival, causal learning, language learning (in humans) injury recovery and avoidance, adaptation to non-stationary environments, and the building of world models, strategic planning, and the avoidance of deceptive rewards."
>
> > How does this framework fit with the existing literature on exploration that considers regret and PAC-MDP results? A discussion on that topic would have been appreciated as the manuscript attempts to unify many existing methods. For example can Equation (22) be rewritten in the form of the Equation (13).
>
> We added a discussion of pure exploitation methods to the introduction as part of our general introduction rewrite at the request of another reviewer.
>
> "Another approach altogether is to use pure exploration and deterministically and sequentially samples all actions for a fixed number of steps, selecting the most valuable option only at the end [...]"
>
> "[...] Pure exploration can often be guaranteed to find the optimal/most rewarding action in bandit settings but is likewise inefficient with large problems. Perhaps more important though is the fact that pure exploration ignores rewards value during its search completely."
>
> > In the experiment section why only consider bandits problems when the paper is about exploration exploitation in reinforcement learning?
>
> We feel bandit tasks were an ideal test bed to focus on studying the exploration-exploitation problem in isolation due their long use in just this setting. Perhaps the reviewer can explain their hesitations in greater detail. I feel I am missing something in your concern.
>
> > [x] Figures 4 and 5 are really small and hard to read.
>
> We use the size of several figures to improve their readability.
>
> ## Required
>
> > Highlight more the relationship between sections 2.2 to 2.3.1 and the rest of manuscript.[x]
>
> Please see our reply above.

---

> > ### Author Response · Authors · 2023-07-05
> > **Review of Paper1070 by Reviewer Crq3 [continued]**
> >
> >
> > > Provide more explanations why equation (14)-(15) is not / could not be directly solved and the WSLS strategy is considered instead.
> >
> > We took the WSLS approach because the expected value of $E$, $V_E$ is non-stationary as $E$ is contracting to 0 during learning. The only way to then estimate the expected value of $E$ for all t in the out horizon $T$ it to run repeated trials. This is both computationally expensive and so often impractical in today's deep RL applications (which we are interested in and are addressed in another paper).
> >
> > We derive everything first using expected values for the simple reason of wanting to develop our theory in a general way that is familiar way in an RL audience.
> >
> > > Can conjecture 1 be proven instead of being an hypothesis?
> >
> > The answer is in short, no. We're not sure how to prove it generally. We do discuss how curiosity is certain to search, visit the entire Markov space, and only revisit states for which there is information to be learned. That is, Curious exploration motivated by a contracting value information (to less that $\eta$) value will completely search any finite space efficiently (in terms of the learning) and will only revisit observations about which these is more to learn (those observations for which $E > \eta$).
> >
> > (At the request of another reviewer the "conjectures" are now labelled as "assumptions".)
> >
> > > Small typos:
> >
> > Corrected. Thank you.

---

### Review · Reviewer_7fYs · 2023-05-29

**Summary Of Contributions:**

This paper presents a formalism for dealing with the exploration-exploitation dilemma. The formalism seeks to encompass previous attempts of formalizing/dealing with the dilemma.

The formalism is rooted at four Axioms defining a metric of information for learning:

1. Memory
2. Specificity
3. Scholarship
4. Equilibrium

The formalism also assumes that reward and information are equally important and that "curiosity" is enough for any exploration problems.

The paper also presents a solution to the exploration-exploitation problem given the formalism and the assumptions. The solution is a heuristic where the policy that maximizes reward is invoked if the latest reward observed is larger or equal to the information metric from the previous time step minus a threshold value; the policy that maximizes exploration is invoked otherwise.

Experiments on multi-armed bandit problems evaluate the heuristic.

**Audience:**

Yes

**Broader Impact Concerns:**

I don't have any broader impact concerns with this submission.

**Claims And Evidence:**

No

**Requested Changes:**

I would expect the authors to deal with the list of weaknesses I wrote above. They can either fix the paper or explain why it has to be done the way it is.

**Strengths And Weaknesses:**

**Strengths**

The paper deals with an important problem and explains in detail the related literature. For example, I appreciated the references from the psychology literature. The paper also explains the theory and insights behind the simple heuristic used to deal with the exploration-exploitation dilemma. Instead of simply showing competitive empirical results, the authors went through the trouble of deriving a theory that attempts to explain why the heuristic is meaningful.

**Weaknesses**

*Major*

The paper is sloppy in some of its definitions, which makes me wonder if some of the contributions of the paper are correct. I apologize in advance to the authors if I misunderstood some parts of the paper. I will try to detail as much as possible the parts I found confusing so the authors can comment on them.

1. I found the definition of $f$ and $f^{-1}$ to be confusing. Function $f$ is the "learning function" that takes an observation and the memory $M$ and returns a new state of memory. This definition is fine, although it would be easier to understand if memory is defined as the parameters of the model and $f$ is an update function. The issue I have is with the "forgetting function" that uses the inverse of $f$. Shouldn't the forgetting function be a function $f'$ that has the same form of $f$ in the sense that it takes an observation and a set of parameters and returns a modified set of parameters $M$.

2. As I hinted above, I also found the definition is of $M$ to be confusing. The paper defines it as a vector of size $p$, but Axiom 2 refers to its elements as $|\Delta M_i|$. Perhaps what Axiom 2 is referring to is to the vector given by $|M_t - M_{t-1}|$, where $M_t$ and $M_{t-1}$ are my own definitions for the memory vectors at time steps $t$ and $t_1$, respectively. What is $M$? Can I think of it as the weights of a neural network or should I think of it as the weighted gradient for a given observation $X$?

3. I suspect that Axiom 2 wasn't worded the way the authors wanted. If states:

> If all $i$ elements $|\Delta M_i|$ in $\boldsymbol{M}$ are equal, then $E$ is minimized.

Unless I am missing something, this Axiom cannot be true. Take Figure 2(a) as an example. $E$ can be very large for two points on the line $M_i = M_j$. For example, for $t$ we have that $M_i = M_j = 0$ and for $t + 1$ we have that $M_i = M_j = 10^{89}$. The value of $E$ is very large despite all elements of $\boldsymbol{M}$ being equal (if we are talking about the difference of $M$-values in $t$ and $t+1$, then they are both $10^{89}$).

4. The notion of consistency can only be defined for a time limit. Take Figure 2(c) as an example. All curves could converge to a value less than $\eta$ after more time steps; it just isn't shown in the plot.

5. I don't see why the following statement is true.

> It also follows from Axiom 3 and 4 that curiosity-driven exploration will visit every state in S at least once in a finite time T.

I don't see the connection between $E \geq 0$ (Axiom 3) and learning "reaching an equilibrium" (Axiom 4) with the guarantee of visiting all states at least once.

6. It isn't clear what notion of equilibrium is used in Axiom 4. Also, it isn't clear whether the same observation $X_i$ is observed over and over again.

7. I don't understand Equation 4. $M$ is defined as a vector but used as a function in Equation 4.

8. Conjecture 1 and 2 are actually assumptions.

9. The formalism presented considers reinforcement learning problems, but the experiments are only on bandits.

10. Why use $\max(E_t, R_t)$ if the assumption is that reward and information collection are equally important? Wouldn't $E_t + R_t$ make more sense? For example, if $E_t$ and $R_t$ are equally important and $o_1 = (E_t = 10, R_t = 9)$ and $o_2 = (E_t = 11, R_t = -90000)$, which one would you prefer? I would go with $o_1$, but the joint value function prefers $o_2$ because it cares about the max.

*Minor*

1. The citations would read better in the format Gupta et al. (2006) instead of Gupta et al. (Gupta et al. 2006).
2. $St'$ should probably be $S'$ (on page 4).
3. I find Equation 1 confusing because $\pi_R$ isn't used anywhere
4. What is the expectation over in Equation 2?
5. "by by" under Axiom 3
6. "Plasticity" wasn't defined.

---

> ### Author Response · Authors · 2023-07-05
> **Re: Review of Paper1070 by Reviewer 7fYs**
>
> Re reviewer had 10 major points of concern about the definitions and notation. We found each to be valid. We appreciate their efforts in particular.  These 10 points are addressed in detail below.
>
> 1. The notation was changed to $f'$. Inverse notation was not a good choice.
> 2. Your interpretation is correct. We rewrote the relevant text to read, "Though we will more often use time indexing $M_{t}$ to denote the updated memory instead. In this notation the difference between and two memories $M_{t-\tau}$ and $M_{t}$ is denoted by $\Delta M$, for some finite time difference $\tau > 0$. We will also need to define a forgetting function, $f'(X_{t-1},\ M_{t}) \rightarrow \mathcal{M}_{t-1}$ which is important later on when establishing our novel Bellman result."
> 3. It was easy to misread from an optimization point of view. We meant the $E$ will the smallest value and any given $\sum \Delta \mathcal{M}$. That is if say $\sum \Delta \mathcal{M} = 0.25$ composed a vector $[0.125, 0.125]$ will have the smallest $E$ compared to any other vector that satisfies that same sum. The updated text reads, "[...]In other words, information about $\mathcal{X}$ that is uniformly encoded in the space of $\mathcal{M}$ is non-specific, or, in a sense, maximum entropy.  We do not mean that when all $i$ elements $\Delta M_i$ in $\Delta \mathcal{M}$ are equal, then $E=0$ and so $E$ is minimized on an absolute sense. Instead we think of this axiom as an ``inverse of entropy'' axiom. Entropy is maximized under a uniform distribution, and so we reason information value is minimized under the same condition. Learned information evenly in memory cannot induce any specific inductive bias on behavior, and is therefore regarded as the least valuable information possible (see also\citep{Mitchell1980}). As an example, if say the two element memory has the $\sum \Delta \mathcal{M} = 0.25$ but is also the vector $[0.125, 0.125]$ this vector will have the smallest $E$ compared to any other vector that satisfies that same sum."
> 4. We agree are were trying to convery this. We rewrote the Axiom to better explain, "Given some observation $\mathcal{X}$, $E$ will decrease below some finite threshold $\eta > 0$ in a finite time horizon $T$."
> 5. We tried to better explain our reasoning in revised paper, "It also follows from Axiom 3 and 4 that curiosity-driven exploration will visit every state in $\mathcal{S}$ at least once in a finite time $T$, assuming $E_0 > 0$. This follows from the fact that once $E - \eta = 0$, exploration of the corresponding observation $\mathcal{X}$ will cease. This implies that each state will eventually become available, forcing the algorithm to visit some other state. This process will repeat until all states have been visited."
> 6. We meant equilibrium loosely, as in unchanging in time. We remove the index from the definition in the axiom as it was confusing as the reviewer noted.
> 7. It should not have been. This was a typo / a hold over from a previous draft where we defined things a differently.
> 8. They are also assumptions. We changed the term.
> 9. This was brought up by another reviewer. We are happy to explain or engage on this point. But we noted there we feel bandit tasks were an ideal test bed to focus on studying the exploration-exploitation problem in isolation due there long use in just this setting. Perhaps the reviewer can explain their hesitations in greater detail. I feel I am missing something in the your concern. Our scheme works for any RL any algorithm we are aware of because the reward part of RL learning is intact and unmodified.
> 10. The are two reasons. First, there is mounting evidence in the neuroscience and behavioral science literature that monkeys, rats, octopi, adult humans, and human kids, often explore in reward-learning settings as if they were maximizing information value. This suggests an independence of exploration. We wanted to explore this idea by putting pure exploration and pure exploitation at odds, defining our game. Joining E + R destroys the very independence we wanted to study. Second, the E + R form introduces a superposition problem. Imagine there are three different tuples denoting (E, R) pair of values. One tuple could be (1, 0), one could be (0, 1), and the final one could be (0.5, 0.5). All these sum to 1, and so are undecidable using E + R. However, we feel from the explore-exploit perspective these three tuples represent very different cases we should distinguish. Our scheme does so in what seems a sensible way, choosing to explore in the first case, and exploit in the other two.

---

> > ### Author Response · Authors · 2023-07-05
> > **Re: Review of Paper1070 by Reviewer 7fYs [continued]**
> >
> > Other minor points:
> >
> >  The citations would read better in the format Gupta et al. (2006) instead of Gupta et al. (Gupta et al. 2006).
> >
> > > St′should probably be S′(on page 4).
> >
> > Thank you.
> >
> > > I find Equation 1 confusing because πR isn't used anywhere
> >
> > We have tried to better explain this derivation by adding the below text to the opening of this section,
> >
> > "We begin by restating the standard reinforcement learning problem of maximizing rewards \citep{Sutton2018}. We do this to set the stage for our derivation for information value and also to define the reward maximizing policy $\pi_R$ used in out final equations (For example, Eq. 15)."
> >
> > > What is the expectation over in Equation 2?
> >
> > > "by by" under Axiom 3
> >
> > Thank you.
> >
> > > "Plasticity" wasn't defined.
> >
> > Plasticity is term common in neuroscience but not so in the ML literature. We removed it and rewrote the surrounding text.

---

### Review · Reviewer_Yatn · 2023-05-31

**Summary Of Contributions:**

The paper aims to solve the problem of exploration and exploitation via a framework that tries to unify both problems. The key idea seems to make the curiosity measurable, via a special value function that measures information with entropy like metrics, and a special “value function” (\pi_ER in the paper) that compares information vs. reward, and decide to explore or exploit based on the comparison.












**Audience:**

Yes

**Claims And Evidence:**

Yes

**Requested Changes:**

thouroughly proof read, read, read your own paper and see if it's understandable to you without recalling much knowledge.

The introduction needs completely rewritten. Currently it is hard to undersand the main idea and theme is not clear.

**Strengths And Weaknesses:**

The formulation of exploration and exploitation via this special structure seems interesting, and Bellman equation for this mode switching can be defined. It appears novel to me.

However, the paper has many writing and presentation problems that i hope the authors take care of it before publishing.

The paper is loosely written and quite hard to follow for me. My background is RL for many years. I’ve worked on exploration in RL but not curiosity. So I’m largely knowledgeable about it but not a super expert.

I guess my main problem is after reading the introduction, I haven’t figured out the key idea and advantages of the paper. What is the relationship between curiosity and uncertainty? There are just a lot of conceptual discussions that are loosely glued together.

The abstract is well written. However, the introduction is very confusing, especially the beginning.

and exploitation is defined as choosing the most valuable action. — this is ambiguous.
t exploration is an essential if risky means to accomplish reward maximization: not very easy to understand this, especially followed with this number of references afterwards.
The risk inherent in exploration is that reward value may be lost in pursuing new options: what do options mean here?

trade=of->tradeoff

I don’t quite understand fig 1. Does it make the exploration and exploitation dilemma easy to solve?

They do so even when that information is costly in terms of time, energy, or both (Song et al., 2019; Wang & Hayden, 2019; Taylor, 1975; Singh, 1970): I found this style of reference is across paper. There is not much information provided about what these reference do except purely pointers. This is also in the beginning of introduction, swarmed with a huge number of references.

Prior work has shown that curiosity leads to the building of intuitive physics (Laversanne-Finot et al., 2018), and may be key to understanding causality (Sontakke et al., 2020).
This is a good style.

In fact, one could conjecture that curiosity is sufficient for all exploration in reinforcement learning (Groth et al., 2021; Fister et al., 2019; Inglis et al., 2001).
This is a pretty strong statement. What evidence are there in these paper to support this?

gradient-based artificial learning systems: a bit strange but interesting.

The contributions in the list do not connect much to the introduction, with new things mentioned that are not defined, e.g., “novel idea of targeted forgetting”, “meta-Bellman””, “win-stay lose-switch” (just mentioned a few references without even briefly mentioning what it is.

local minima in the reward value function: here it’s too ambiguous. Do you mean the reward function? The value function? By local minima, what exactly is it? Local minima is a loss or objective. For RL, it is maximum.


“Instead we aim to develop a mathematical account that generalizes across all of them”
This seems ambitious and nice. But shouldn’t we at least discuss one or two typical such cited works in detail before getting into this goal?

the learning progress literature (Kaplan & Oudeyer, 2007; Lopes et al., 2012; Baranes & Oudeyer, 2013; Ten et al., 2021). The key point of difference between our approach and this prior work is that we do not include in our value metric the …
Here I found it’s confusing again. You introduced a concept “learning progress” and a few references. Before describing it or any of them, you started “the difference” of your work…
I see you defined this in Sec 2.2.1. The discussion in this section about memory is catchy.

If all i elements |∆Mi | in M are equal: what exactly does this mean? What is “information that is learned evenly in memory ”? Like a tie in the Q value function? But there are sometimes some problems like this. Two actions are equally important, we can choose either of them. Then this Q is not favourable? I’m lost here.

Curiosity is a sufficient solution for all exploration problems (where learning is possible). I found it’s hard to judge on this.

Eq. 5 just doesn’t look comfortably. Where is a in E?

I understand your approach finally seems like hierarchical thing from eq. 12, which compares the information vs. reward. So I guess this and together 13 are the key ideas of the paper.

---

> ### Author Response · Authors · 2023-07-05
> **Re: Review of Paper1070 by Reviewer Yatn**
>
> We reworked and rewrote the introduction to try and answer your questions. In doing so we addressed the overall style of the introductions, and filled in conceptual gaps. Detailed replies follow, spread over several comments due to open review limits.
>
> > I don’t quite understand fig 1. Does it make the exploration and exploitation dilemma easy to solve?
>
> To try and explain for all readers, we adding the following to Fig 1, "[...] In the dilemma (a.) the expected value of exploration is uncertain. Wheres to solve our alternative view (b.), we set information and reward value on equal terms, approximate their expected values, and derive a deterministic, optimal value rule to choose between exploration and exploitation. In other words, or alternative transforms the fundamental explore-exploit dilemma into a simple greedy decision."
>
> > They do so even when that information is costly in terms of time, energy, or both (Song et al., 2019; Wang & Hayden, 2019; Taylor, 1975; Singh, 1970): I found this style of reference is across paper. There is not much information provided about what these reference do except purely pointers. This is also in the beginning of introduction, swarmed with a huge number of references. Prior work has shown that curiosity leads to the building of intuitive physics (Laversanne-Finot et al., 2018), and may be key to understanding causality (Sontakke et al., 2020). This is a good style. [x]
> >
> > In fact, one could conjecture that curiosity is sufficient for all exploration in reinforcement learning (Groth et al., 2021; Fister et al., 2019; Inglis et al., 2001). This is a pretty strong statement. What evidence are there in these paper to support this?
>
> The large number of references and the discussion of the universality of curiosity in the animal world was intended to bolster our case for out curious conjecture. We tried to make this connection and case more explicit in the introduction, "One common strategy, $\epsilon$-greedy \cite{Sutton2018}, handles both uncertainties by randomly make both decisions. A more directed approach is to augment reward values from the environment with intrinsic rewards or motivations. A variety of intrinsic motivation strategies are available in the literature. Examples include, novelty signals \cite{Kakade2002}, action counts \cite{Bellemare2016}, information gain \cite{Friston2017}, error maps \cite{Thrun1992}, curiosity \cite{Schmidhuber1991} and learning progress \cite{Kaplan2007}. Another approach altogether is to use pure exploration and deterministically and sequentially samples all actions for a fixed number of steps, selecting the most valuable option only at the end \cite{Brafman2002,Strehl,Kearns2002}."
>
> "Randomly resolving the dilemma as in $\epsilon$-greedy is effective and in fact a common solution in the literature. It is however inefficient and can struggle in continuous and high-dimensional action/state spaces \citep{Sutton2018}. Pure exploration can often be guaranteed to find the optimal/most rewarding action in bandit settings but is likewise inefficient with large problems \citep{Brafman2002,Seldin}. Perhaps more important is that pure exploration ignores rewards value during its search completely. Whereas exploration using intrinsic rewards will often fair better for larger problems, but requires parameter tuning and so can be computationally expensive to arrive optimal explore-exploit strategies \cite{Guez2013,Asmuth2009}."

---

> > ### Author Response · Authors · 2023-07-05
> > **Re: Review of Paper1070 by Reviewer Yatn [continued]**
> >
> >
> > > The contributions in the list do not connect much to the introduction, with new things mentioned that are not defined, e.g., “novel idea of targeted forgetting”, “meta-Bellman””, “win-stay lose-switch” (just mentioned a few references without even briefly mentioning what it is. [x]
> >
> > We rewrote the introduction and the contributions list to more tightly reference each other better. Please see the update paper.
> >
> > > local minima in the reward value function: here it’s too ambiguous. Do you mean the reward function? The value function? By local minima, what exactly is it? Local minima is a loss or objective. For RL, it is maximum. [x]
> >
> > The term local minima was misleading. So we removed it. We instead describe "deceptive value functions", which we now define explicitly as, "[...] It can ensure deceptive value functions, those whose rewards appear suboptimal early in learning but are optimal in the long term, are overcome during learning \citep{Fister2019,Pathak2019}. This is an especially important point that we revisit in our experiments later on."
> >
> > > “Instead we aim to develop a mathematical account that generalizes across all of them” This seems ambitious and nice. But shouldn’t we at least discuss one or two typical such cited works in detail before getting into this goal?
> >
> > We added a detailed discussion of other explorations methods, and discussed what exactly we propose to generalize. Please see our reply above for the details.
> >
> > > the learning progress literature (Kaplan & Oudeyer, 2007; Lopes et al., 2012; Baranes & Oudeyer, 2013; Ten et al., 2021). The key point of difference between our approach and this prior work is that we do not include in our value metric the … Here I found it’s confusing again. You introduced a concept “learning progress” and a few references. Before describing it or any of them, you started “the difference” of your work… I see you defined this in Sec 2.2.1. The discussion in this section about memory is catchy. [x]
> >
> > We added more concrete motivation and background for our work on learning progress.
> >
> > "However despite curiosity being a powerful method to build world models, avoid local minima, discover causation, there remain two open problems in curiosity research we felt needed to be addressed before curiosity could be fully incorporated into reinforcement learning theory in a general way:
> >
> > "The existing definitions of curiosity and information value are either task dependent, memory specific, or specific to the learning rule. Or all of the above. For example, in a recent review Oudeyer (\citep{Oudeyer2007} created a formal typology of computational theories of intrinsic motivation during learning, focusing on the creating classifications of intrinsic motivation during learning based on in part on their motivations or goals. He stops short though of defining a generalized metric, which is what we offer here."
> >
> > "[...]Curiosity algorithms in artificial systems suffer from a "white noise" problem where they become fixated on ``useless'' high entropy features in the environment \citep{Pathak2017,Kim2020}."
> >
> > > If all i elements |∆Mi | in M are equal: what exactly does this mean? What is “information that is learned evenly in memory ”? Like a tie in the Q value function? But there are sometimes some problems like this. Two actions are equally important, we can choose either of them. Then this Q is not favourable? I’m lost here.
> >
> > There was confusion in two of the reviewers about this axiom. We added the below the better explain:
> >
> > "In other words, information about $\mathcal{X}$ that is uniformly encoded in the space of $\mathcal{M}$ is non-specific, or, in a sense, maximum entropy.  We do not mean that when all $i$ elements $\Delta M_i$ in $\Delta \mathcal{M}$ are equal, then $E=0$ and so $E$ is minimized on an absolute sense. Instead we think of this axiom as an inverse of entropy axiom. Entropy is maximized under a uniform distribution, and so we reason information value is minimized under the same condition. Learned information evenly in memory cannot induce any specific inductive bias on behavior, and is therefore regarded as the least valuable information possible (see also\citep{Mitchell1980}). As an example, if say the two element memory difference with the $\sum \Delta \mathcal{M} = 0.25$ but is also the vector $[0.125, 0.125]$ then this vector will have the smallest $E$ compared to any other vector that satisfies that same sum."

---

> > > ### Author Response · Authors · 2023-07-05
> > > **Re: Review of Paper1070 by Reviewer Yatn [continued]**
> > >
> > > > Curiosity is a sufficient solution for all exploration problems (where learning is possible). I found it’s hard to judge on this.
> > >
> > > We have tried to better explain our motivations in the introduction. In brief, in the animal kingdom curiosity is universally observed behavior that is often prioritized over rewards collection. Second, algorithmic studies of artificial curiosity show it is generally effective and has diverse benefits not found in objective-based search. We note artificial curiosity has one significant downside--a tendency fixate on the "white noise" stimuli. (An issue we resolve fully in this work).
> > >
> > > > Eq. 5 just doesn’t look comfortably. Where is a in E?
> > >
> > > These equations, 5-11 were carry overs from the older draft and should not have been present. They were not, as you note, mathematically sensible. They were degenerate with the earlier presentation.
> > >
> > > > I understand your approach finally seems like hierarchical thing from eq. 12, which compares the information vs. reward. So I guess this and together 13 are the key ideas of the paper.
> > >
> > > This is exactly it.

---

### Decision · Action_Editors · 2023-07-23

**Recommendation:** Reject

**Comment:**

All three reviewers favored rejecting this submission.

While reviewer Yatn understood the high-level motivation of the paper, they indicated in their review that they found the paper difficult to follow, and they did not find the core contributions sufficiently accessible.  Their final recommendation was to reject the paper, in part due to readability.

Reviewer 7fYs also understood the high-level motivation of the paper and believes the problem is important. This reviewer also engaged with the paper in a detailed fashion, identifying numerous points, including raising potential issues with the theoretical analysis.  The authors responded to this review indicating that the points raised by the reviewer were valid and that they have attempted to address them.  This reviewer recommended that the paper be rejected before the authors responded (due to a delayed response by the authors) and this reviewer did not change their recommendation after the author response was posted.

Reviewer Crq3 also appreciated the high-level themes in the paper, but found the paper difficult to follow.  The authors attempted to address some of the specific parts of the text highlighted in the review.  However, this reviewer also made a final recommendation to reject the paper (after seeing the author responses).

A fair summary overall is that the current form of the paper is not sufficiently accessible to this set of reviewers, making evaluation of the merits of the specific claims difficult.  Given that I have confidence in this set of reviewers as qualified to evaluate this work, I am obligated to reject this submission, in its current form.  My recommendation to the authors would be to streamline the presentation and make the core ideas more accessible.


**Audience:**

I think the paper topic is generally appropriate for this venue.

**Claims And Evidence:**

According to the reviewers, this paper does not pass the bar for "claims and evidence".  All of the reviewers struggled in terms of following the presentation of the paper.  Reviewer 7fYs, who engaged most substantively with the theoretical analysis, pointed to specific details which were presented in a way that was confusing or otherwise difficult for them to evaluate.  I do not have confidence that any of the reviewers were able to fully understand or evaluate the claims presented in the paper (however, I'd emphasize that I believe the reviewer selection is reasonable and representative of the kinds of expert researchers who would be expected to read this paper).

**Resubmission Of Major Revision:**

The authors may consider submitting a major revision at a later time.